# Lossless Vocabulary Reduction for Auto-Regressive Language Models

**Daiki Chijiwa**[1*]**, Taku Hasegawa**[2]**, Kyosuke Nishida**[2]**, Shin'ya Yamaguchi**[1]**,
Tomoya Ohba**[1]**, Tamao Sakao**[1]**, Susumu Takeuchi**[1]

[1]NTT Computer and Data Science Laboratories, NTT Corporation
[2]NTT Human Informatics Laboratories, NTT Corporation

## Abstract

Tokenization—the process of decomposing a given text into a sequence of subwords called *tokens*—is one of the key components in the development of language models. Particularly, auto-regressive language models generate texts token by token, i.e., by predicting the next-token distribution given the previous ones, and thus tokenization directly affects their efficiency in text generation. Since each language model has their own vocabulary as a set of possible tokens, they struggle to cooperate with each other at the level of next-token distributions such as model ensemble. In this paper, we establish a theoretical framework of lossless vocabulary reduction, which efficiently converts a given auto-regressive language model into the one with an arbitrarily small vocabulary without any loss in accuracy. This framework allows language models with different tokenization to cooperate with each other efficiently by reduction to their maximal common vocabulary. Specifically, we empirically demonstrate its applicability to model ensemble with different tokenization.

## 1 Introduction

Tokenization—the process of decomposing a given text into a sequence of subwords called *tokens*—plays an important role in modern language models (Schuster & Nakajima, 2012; Sennrich et al., 2016; Kudo, 2018), where tokens are the minimum unit for their input and output. Particularly, tokenization largely affects the efficiency of text generation with auto-regressive language models (Radford et al., 2019) which are trained to generate texts by first computing the next-token distribution given previous tokens and then sampling a token from it iteratively. In other words, the longer tokens are sampled at each iteration on average, the less number of sampling iterations is required for text generation.

Each language model has its own *vocabulary*, the set of all possible tokens, which is generally constructed based on the statistics in their training data so that more plausible texts can be represented by less numbers of tokens. As a result, given two (or more) language models that have been trained independently with distinct training data, their vocabularies do not match in general. Due to the vocabulary mismatch, *language models with different tokenizers or vocabularies struggle to cooperate with each other at the level of their next-token distributions*, such as ensemble (Hinton, 1999), knowledge distillation (Hinton et al., 2015), speculative decoding (Leviathan et al., 2023), inference-time alignment (Mitchell et al., 2024), etc.

To this problem, recent work (Phan et al., 2025; Vieira et al., 2025) have proposed a theoretically-guaranteed approach that reduces a given next-token distribution to the corresponding next-*byte* distribution, without changing the distribution of generated texts. In other words, the resulting byte-level distribution is equivalent to the original token-level one as a probabilistic text generator, while its vocabulary being restricted to the set of all one-byte tokens, $\langle 0x00 \rangle$ to $\langle 0xFF \rangle$. This approach enables language models with different vocabularies to cooperate with each other, at the level of their next-byte distributions. However, the byte-level cooperation leads to increased inference costs by its nature, since each model has to predict byte by byte instead of tokens with multiple bytes.

In this paper, beyond the byte-level reduction, we establish the first theoretical framework called **lossless vocabulary reduction** that reduces a given next-token distribution to the corresponding one over an *arbitrary sub-vocabulary* without changing its behavior as a text generator (Figure 1),

---

*Correspondence to: `daiki.chijiwa@ntt.com`

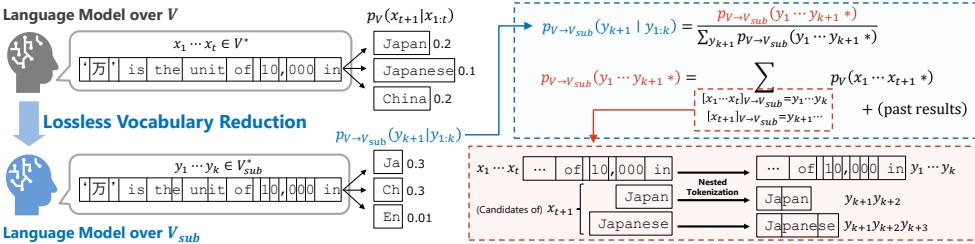

Figure 1: Overview of lossless vocabulary reduction. Instead of sampling tokens from the original next-token distribution over $\mathcal{V}$, we can inductively compute and sample from the equivalent distribution over the sub-vocabulary $\mathcal{V}_{\text{sub}}$ while keeping its accuracy. See Section 3 for notations and details.

which is achieved by introducing the new notion of **nested tokenization**. Then we derive an efficient approximated algorithm to compute it with negligibly small overhead. As an application, we propose to cooperate language models with different vocabularies by lossless reduction to their **maximal common vocabulary** among them.

Finally, our contributions in this paper can be summarized as follows:

1. We established the first theoretical framework of lossless vocabulary reduction and derived an efficient approximated algorithm that converts given next-token distributions over some vocabulary to the equivalent distributions over its arbitrary sub-vocabulary in inference-time. We also provided an illustrative example of how it actually works following our theory.

2. Experimental results with several language models show that the derived algorithm is actually almost lossless as our theory suggests. Also, experimental results on ensemble show that ensemble over the maximal common vocabulary achieves comparable accuracy to the byte-level ensemble while the former is more efficient than the latter.

## 2 PRELIMINARIES

Throughout this paper, we assume that any text is a sequence of bytes obtained by some character encoding, typically by UTF-8 (Yergeau, 2003). In this section, we briefly introduce the formal definitions and properties of texts, tokenization, and language models.

### 2.1 FORMULATION OF TOKENS

**Texts.** Let $\mathcal{A}$ be a set of symbols, such as all alphabets or all characters. Throughout this paper, $\mathcal{A}$ is considered as a set of all bytes, i.e., 8-bit strings $b_1 \cdots b_8$ with $b_i \in \{0, 1\}$. Let $\mathcal{A}^* := \bigcup_{k=1}^{\infty} A^k \cup \{\emptyset\} = \{a_1 \cdots a_N \mid a_i \in \mathcal{A}, N \in \mathbb{N}\}$[1] be the set of all finite sequences of bytes with an empty symbol $\emptyset$. We often use the notation $a_{1:N} := a_1 \cdots a_N$ for simplicity. Here we briefly note that (1) $\mathcal{A}$ consists of only 256 elements, obviously less than all symbols in the real world, (2) $\mathcal{A}^*$ consists of all possible texts in computers because they are represented by sequences of bytes.

**Tokenization.** A tokenization scheme $\mathcal{T}$ is formally defined[2] as a triplet $(\mathcal{V}, [-]_{\mathcal{V}}, [-]_{\mathcal{A}})$, with a finite set of vocabulary $\mathcal{V}$, an encoder $[-]_{\mathcal{V}} : \mathcal{A}^* \to \mathcal{V}^*$ and a decoder $[-]_{\mathcal{A}} : \mathcal{V}^* \to \mathcal{A}^*$ satisfying:

$$[[a_{1:N}]_{\mathcal{V}}]_{\mathcal{A}} = a_{1:N} \text{ for all texts } a_{1:N} \in \mathcal{A}^*, \text{ and} \tag{1}$$
$$[x_{1:T}]_{\mathcal{A}} = [x_1]_{\mathcal{A}} \cdots [x_T]_{\mathcal{A}} \text{ for all tokens } x_{1:T} \in \mathcal{V}^*.$$

Each $x \in \mathcal{V}$ is called a token in $\mathcal{T}$, and has its textual representation $[x]_{\mathcal{A}} \in \mathcal{A}^*$ given by the decoder. The encoder $[-]_{\mathcal{V}}$ uniquely converts a given text $a_{1:N} \in \mathcal{A}^*$ to the corresponding tokens $x_{1:T} = [a_{1:N}]_{\mathcal{V}}$ with some length $T$. Note that the condition (1) has first appeared in Kudo & Richardson (2018) by treating texts as a sequence of Unicode bytes such as UTF-8 (Yergeau, 2003), which is now employed by most of modern language models after Radford et al. (2019).

---

[1]The construction of $\mathcal{A}^*$ is also known as the Kleene closure of $\mathcal{A}$, and $\emptyset$ means the empty string.
[2]Throughout this paper, we focus only on deterministic tokenization which is widely employed in modern language models.

## 2.2 FORMULATION AND PROPERTIES OF LANGUAGE MODELS

**Language model over tokens.** Let $p_{\text{text}}(a_{1:N})$ be the true distribution of natural language texts, defined over sequences of 8-bit binaries $a_{1:N} \in \mathcal{A}^*$. A language model $p_{\mathcal{V}}(x_{1:T})$, with respect to the tokenization $\mathcal{T} = (\mathcal{V}, [-]_{\mathcal{V}}, [-]_{\mathcal{A}})$, is defined as the probabilistic distribution over sequences of tokens $x_{1:T} \in \mathcal{V}^*$ that matches the distribution of the encoded sequences $[a_{1:N}]_{\mathcal{V}}$ of natural texts $a_{1:N} \sim p_{\text{text}}(a_{1:N})$. In other words, it is defined as

$$p_{\mathcal{V}}(x_{1:T}) := \begin{cases} p_{\text{text}}(a_{1:N}), & \text{if } x_{1:T} = [a_{1:N}]_{\mathcal{V}} \text{ for some (unique) } a_{1:N} \in \mathcal{A}^*, \\ 0, & \text{otherwise.} \end{cases} \tag{2}$$

Then we say that $p_{\mathcal{V}}(x_{1:T})$ is a token distribution associated with the true text distribution $p_{\text{text}}(a_{1:N})$.

**Next-token distribution.** From a computational perspective, it is infeasible to implement the token distribution $p_{\mathcal{V}}(x_{1:T})$ directly. Rather, it is standard to implement its *next-token distribution* denoted as $p_{\mathcal{V}}(x_t \mid x_{1:t-1})$, which we formally define here. First of all, let us introduce $x_1 \cdots x_t *$, or $x_{1:t}*$ in shorthand, a set of all tokens starting with $x_1 \cdots x_t$:

$$x_1 \cdots x_t * := \{x_1 \cdots x_t x_{t+1} \cdots x_T \in \mathcal{V}^* \mid x_{t+1:T} \in \mathcal{V}^*, T \in \mathbb{N}\} \subset \mathcal{V}^*, \tag{3}$$

which plays a central role here. We can consider the corresponding probability to this event:

$$p_{\mathcal{V}}(x_1 \cdots x_t *) = \sum_{x_{1:T} \in x_1 \cdots x_t *} p_{\mathcal{V}}(x_1 \cdots x_t x_{t+1} \cdots x_T) \tag{4}$$

Now the next-token distribution $p_{\mathcal{V}}(x_t \mid x_{1:t-1})$ is defined as the conditional probability

$$p_{\mathcal{V}}(x_t \mid x_{1:t-1}) := p_{\mathcal{V}}(x_{1:t}* \mid x_{1:t-1}*) = \frac{p_{\mathcal{V}}(x_{1:t}*)}{p_{\mathcal{V}}(x_{1:t-1}*)}. \tag{5}$$

We can sample a full sequence $x_{1:T} \in \mathcal{V}^*$ from the distribution $p_{\mathcal{V}}(x_{1:T})$ by recursively sampling $x_t$ from the next-token distribution $p_{\mathcal{V}}(x_t \mid x_{1:t-1})$ starting from the empty string $x_0 = \emptyset$. Hence, it is sufficient to implement the next-token distribution $p_{\mathcal{V}}(x_t \mid x_{1:t-1})$ for the task of sequential text generation from $p_{\mathcal{V}}(x_{1:T})$. Throughout this paper, we assume that the next-token distribution $p_{\mathcal{V}}(x_t \mid x_{1:t-1})$ can be computed for all $x_t \in \mathcal{V}$ in parallel by a single unit of computation, as an output of a neural network followed by the softmax layer over the vocabulary $\mathcal{V}$.

**Valid tokens and minimal covering.** Let $x_{1:T} \in \mathcal{V}^*$ be a sequence of tokens. We call $x_{1:T}$ is a *valid tokens* if it satisfies the following inverse relation of Equation (1):

$$[[x_{1:T}]_{\mathcal{A}}]_{\mathcal{V}} = x_{1:T}. \tag{6}$$

In other words, $x_{1:T}$ is valid if and only if it can be obtained by tokenizing some text $a_{1:N}$, i.e., $x_{1:T} = [a_{1:N}]_{\mathcal{V}}$. Indeed, in the latter case, we can see $x_{1:T}$ is valid since $[[x_{1:T}]_{\mathcal{A}}]_{\mathcal{V}} = [[[a_{1:N}]_{\mathcal{V}}]_{\mathcal{A}}]_{\mathcal{V}} = [a_{1:N}]_{\mathcal{V}} = x_{1:T}$ by Equation (1). Obviously, if $x_{1:T}$ is a valid tokens, it is obtained by tokenizing $a_{1:N} := [x_{1:T}]_{\mathcal{A}}$, the stringification of $x_{1:T}$, clearly followed by Equation (6). Note that Equation (6) generally does not hold solely by Equation (1), and actually most sequences of tokens are invalid (Phan et al., 2024).

It is easy to see that the language model $p_{\mathcal{V}}(x_{1:T})$ associated with the true text distribution $p_{\text{text}}(a_{1:N})$ satisfies the following *validity condition* (Phan et al., 2024):

$$p_{\mathcal{V}}(x_{1:T}) = 0, \quad \text{for any invalid tokens } x_{1:T} \in \mathcal{V}^*. \tag{7}$$

Indeed, by the above discussion, the invalid tokens $x_{1:T}$ is never obtained by tokenizing any text $a_{1:N}$. Thus the definition of $p_{\mathcal{V}}(x_{1:T})$ falls into the second case in Equation (2).

Importantly, the validity condition leads to simplification of probability computation for language models. To explain it, let us define the minimal cover $C_{\mathcal{V}}(a_{1:N})$ for given texts $a_{1:N}$ by

$$C_{\mathcal{V}}(a_{1:N}) := \{x_{1:T} \in \mathcal{V}^* \mid a_{1:N} \not\prec [x_{1:T-1}]_{\mathcal{A}}, a_{1:N} \prec [x_{1:T}]_{\mathcal{A}}, \text{ and } x_{1:T} \text{ is valid.}\}. \tag{8}$$

Note that this is not an empty set because $[a_{1:N}]_{\mathcal{V}} \in C_{\mathcal{V}}(a_{1:N})$ always holds. Using this notion, Phan et al. (2025) and Vieira et al. (2025) provided the following formula for computing the underlying distribution over alphabets from the language model over tokens:

**Lemma 2.1** (Phan et al. (2025); Vieira et al. (2025)). *Let $\mathcal{V}$ be any vocabulary and $p_{\mathcal{V}}$ be the token distribution associated with the text distribution $p_{\text{text}}$. For any string $a_1 \cdots a_N \in \mathcal{A}^N$, we have*

$$p_{\text{text}}(a_1 \cdots a_N *) = \sum_{x_{1:T} \in C_{\mathcal{V}}(a_{1:N})} p_{\mathcal{V}}(x_1 \cdots x_T *). \tag{9}$$

Since the minimal cover contains only valid tokens, the sum of the right-hand side will be computationally feasible, contrary to the case without the validity condition where the minimal cover may grow exponentially. In particular, based on this formula, Phan et al. (2025) derived an efficient algorithm to compute the next-byte distribution $p_{\text{text}}(a_t \mid a_{1:t-1})$ from the one over tokens.

# 3 LOSSLESS VOCABULARY REDUCTION

In this section, we establish a theory and derive an algorithm to restrict the token vocabulary $\mathcal{V}$ of a language model $p_{\mathcal{V}}(x_{1:T})$ to any given sub-vocabulary $\mathcal{V}_{\text{sub}}$, without changing its behavior as a text generator. The theory and algorithm in this section will be leveraged for cooperation of language models with different vocabularies in the later sections.

## 3.1 FORMULATION OF VOCABULARY REDUCTION

Suppose that we have a language model $p_{\mathcal{V}}(x_{1:T})$, a distribution over tokens in a vocabulary $\mathcal{V}$ associated with the true text distribution $p_{\text{text}}(a_{1:N})$, and denote its tokenizer $\mathcal{T}_{\mathcal{V}} = (\mathcal{V}, [-]_{\mathcal{V}}, [-]_{\mathcal{A}})$.

**Nested tokenization.** Let $\mathcal{V}_{\text{sub}} \subset \mathcal{V}$ be a sub-vocabulary, i.e., an arbitrary subset of the given vocabulary $\mathcal{V}$, and $\mathcal{T}_{\mathcal{V}_{\text{sub}}} = (\mathcal{V}_{\text{sub}}, [-]_{\mathcal{V}_{\text{sub}}}, [-]_{\mathcal{A}})$ be some tokenization scheme with the sub-vocabulary. Here we do not necessarily impose any relation between $[-]_{\mathcal{V}}$ and $[-]_{\mathcal{V}_{\text{sub}}}$ except for the inclusion between the vocabularies.

Given such a tokenization scheme $\mathcal{T}_{\mathcal{V}_{\text{sub}}}$ with the sub-vocabulary $\mathcal{V}_{\text{sub}}$, we can define the *nested tokenization* $\mathcal{T}_{\mathcal{V} \to \mathcal{V}_{\text{sub}}} = (\mathcal{V}_{\text{sub}}, [-]_{\mathcal{V} \to \mathcal{V}_{\text{sub}}}, [-]_{\mathcal{A}})$ that tokenizes text by applying $[-]_{\mathcal{V}}$ and $[-]_{\mathcal{V}_{\text{sub}}}$:

$$[a_{1:N}]_{\mathcal{V} \to \mathcal{V}_{\text{sub}}} := [[a_{1:N}]_{\mathcal{V}}]_{\mathcal{V} \to \mathcal{V}_{\text{sub}}} \text{ for } a_{1:N} \in \mathcal{A}^*, \text{ with} \tag{10}$$

$$[x_{1:T}]_{\mathcal{V} \to \mathcal{V}_{\text{sub}}} := [x_1]_{\mathcal{V} \to \mathcal{V}_{\text{sub}}} \cdots [x_T]_{\mathcal{V} \to \mathcal{V}_{\text{sub}}}, \quad [x_t]_{\mathcal{V} \to \mathcal{V}_{\text{sub}}} := [[x_t]_{\mathcal{A}}]_{\mathcal{V}_{\text{sub}}} \text{ for } x_{1:T} \in \mathcal{V}^*,$$

where we abuse the notation $[-]_{\mathcal{V} \to \mathcal{V}_{\text{sub}}}$ for denoting both $[-]_{\mathcal{V} \to \mathcal{V}_{\text{sub}}} : \mathcal{A}^* \to \mathcal{V}_{\text{sub}}^*$ and $[-]_{\mathcal{V} \to \mathcal{V}_{\text{sub}}} : \mathcal{V}^* \to \mathcal{V}_{\text{sub}}^*$, as well as $[-]_{\mathcal{A}}$, for simplicity. The nested tokenization will play a central role in our vocabulary reduction. If we consider the case of $\mathcal{V}_{\text{sub}} = \mathcal{A}$ and $\mathcal{T}_{\mathcal{A}} = (\mathcal{A}, \text{id}_{\mathcal{A}}, \text{id}_{\mathcal{A}})$, the nested tokenization $\mathcal{T}_{\mathcal{V} \to \mathcal{A}}$ is just the stringification of given tokens.

**Vocabulary reduction.** Here we introduce a new language model $p_{\mathcal{V} \to \mathcal{V}_{\text{sub}}}(y_{1:K})$ over tokens in the sub-vocabulary $\mathcal{V}_{\text{sub}}$, which is induced by the given language model $p_{\mathcal{V}}(x_{1:T})$ and the nested tokenization $\mathcal{T}_{\mathcal{V} \to \mathcal{V}_{\text{sub}}}$ as follows:

$$p_{\mathcal{V} \to \mathcal{V}_{\text{sub}}}(y_{1:K}) := \mathbb{E}_{x_{1:T} \sim p_{\mathcal{V}}(x_{1:T})}[\mathbb{1}\{y_{1:K} = [x_{1:T}]_{\mathcal{V} \to \mathcal{V}_{\text{sub}}}\}] = \mathbb{P}(y_{1:K} = [x_{1:T}]_{\mathcal{V} \to \mathcal{V}_{\text{sub}}}), \tag{11}$$

i.e., the probability that $y_{1:K}$ is obtained as a re-tokenization of $x_{1:T} \in \mathcal{V}^*$ by the sub-vocabulary $\mathcal{V}_{\text{sub}}$. We call $p_{\mathcal{V} \to \mathcal{V}_{\text{sub}}}(y_{1:K})$ the *vocabulary reduction* of $p_{\mathcal{V}}(x_{1:T})$ onto $\mathcal{V}_{\text{sub}}$.

For example, if we employ the above $\mathcal{T}_{\mathcal{V} \to \mathcal{A}}$ as the nested tokenization, the induced distribution $p_{\mathcal{V} \to \mathcal{A}}(a_{1:N})$ is nothing but the text distribution induced by the stringification of tokens $x_{1:T}$ from $p_{\mathcal{V}}(x_{1:T})$. Particularly, if $p_{\mathcal{V}}(x_{1:T})$ is associated with the true text distribution $p_{\text{text}}(a_{1:N})$, we have

$$p_{\mathcal{V} \to \mathcal{A}}(a_{1:N}) = \mathbb{P}(a_{1:N} = [x_{1:T}]_{\mathcal{A}}) = p_{\mathcal{V}}([a_{1:N}]_{\mathcal{V}}) = p_{\text{text}}(a_{1:N}), \tag{12}$$

by Equation (2). Thus our definition of vocabulary reduction involves the previous byte-level reduction (Phan et al., 2025; Vieira et al., 2025) as its special case with $\mathcal{V}_{\text{sub}} = \mathcal{A}$.

Moreover, the vocabulary reduction $p_{\mathcal{V} \to \mathcal{V}_{\text{sub}}}$ is *lossless*. To see this, we consider the induced text distribution $p_{\mathcal{V} \to \mathcal{V}_{\text{sub}} \to \mathcal{A}}(a_{1:N})$, i.e., the text distribution obtained as the vocabulary reduction of $p_{\mathcal{V} \to \mathcal{V}_{\text{sub}}}$ by another nested tokenization $\mathcal{T}_{\mathcal{V}_{\text{sub}} \to \mathcal{A}}$. By iteratively applying Equation (11), we have

$$p_{\mathcal{V} \to \mathcal{V}_{\text{sub}} \to \mathcal{A}}(a_{1:N}) = \mathbb{E}_{y_{1:K} \sim p_{\mathcal{V} \to \mathcal{V}_{\text{sub}}}(y_{1:K})}[\mathbb{1}\{a_{1:N} = [y_{1:K}]_{\mathcal{A}}\}]$$

$$= \mathbb{E}_{x_{1:T} \sim p_{\mathcal{V}}(x_{1:T})}[\mathbb{1}\{a_{1:N} = [[x_{1:T}]_{\mathcal{V} \to \mathcal{V}_{\text{sub}}}]_{\mathcal{A}}\}]$$
$$= \mathbb{E}_{x_{1:T} \sim p_{\mathcal{V}}(x_{1:T})}[\mathbb{1}\{a_{1:N} = [x_{1:T}]_{\mathcal{A}}\}]$$
$$= p_{\mathcal{V} \to \mathcal{A}}(a_{1:N})$$

By combining these equalities, we have proved the lossless property of vocabulary reduction:

**Theorem 3.1.** *Let $p_{\text{text}}$ be the true text distribution underlying the language model $p_{\mathcal{V}}$. Let us denote the text distribution associated with $p_{\mathcal{V} \to \mathcal{V}_{\text{sub}}}$ by $p_{\mathcal{V} \to \mathcal{V}_{\text{sub}} \to \mathcal{A}}(a_{1:N})$. Then we have*

$$p_{\mathcal{V} \to \mathcal{V}_{\text{sub}} \to \mathcal{A}}(a_{1:N}) = p_{\mathcal{V} \to \mathcal{A}}(a_{1:N}) = p_{\text{text}}(a_{1:N}). \tag{13}$$

### 3.2 COMPUTATION OF THE NEXT-TOKEN DISTRIBUTION

At first glance, based on the above formulation, the computation of vocabulary reduction appears to be simple and easy at the level of probabilities over sentences, i.e., over $\mathcal{V}_{\text{sub}}^*$ and $\mathcal{V}^*$. But in reality, the probabilities over sentences are computationally infeasible in the real world. Instead, we have to compute the corresponding next-token distribution $p_{\mathcal{V} \to \mathcal{V}_{\text{sub}}}(y_{k+1} \mid y_{1:k})$ over $\mathcal{V}_{\text{sub}}$ using the original next-token distribution $p_{\mathcal{V}}(x_{t+1} \mid x_{1:t})$ over $\mathcal{V}$. Since the next-token distribution $p_{\mathcal{V} \to \mathcal{V}_{\text{sub}}}(y_{k+1} \mid y_{1:k})$ can be expressed as

$$p_{\mathcal{V} \to \mathcal{V}_{\text{sub}}}(y_{k+1} \mid y_{1:k}) = \frac{p_{\mathcal{V} \to \mathcal{V}_{\text{sub}}}(y_{1:k+1}*)}{p_{\mathcal{V} \to \mathcal{V}_{\text{sub}}}(y_{1:k}*)} = \frac{p_{\mathcal{V} \to \mathcal{V}_{\text{sub}}}(y_{1:k+1}*)}{\sum_{y_{k+1} \in \mathcal{V}_{\text{sub}}} p_{\mathcal{V} \to \mathcal{V}_{\text{sub}}}(y_{1:k+1}*)} \tag{14}$$

by its definition, the problem is how to compute the marginal distributions $p_{\mathcal{V} \to \mathcal{V}_{\text{sub}}}(y_{1:k}*)$. To answer this problem, we introduce the *relative covering* $C_{\mathcal{V}, \mathcal{V}_{\text{sub}}}(y_{1:k})$, generalizing the minimal covering for $\mathcal{V}$ considered in Phan et al. (2025); Vieira et al. (2025), as follows:

$$C_{\mathcal{V}, \mathcal{V}_{\text{sub}}}(y_{1:k}) := \{x_{1:t} \in \mathcal{V}^* \mid x_{1:t} \text{ is valid and satisfies } y_{1:k} \prec [x_{1:t}]_{\mathcal{V} \to \mathcal{V}_{\text{sub}}}$$
$$\text{and } y_{1:k} \not\prec [x_{1:t-1}]_{\mathcal{V} \to \mathcal{V}_{\text{sub}}}\}, \tag{15}$$

**Lemma 3.2.** *Let $p_{\mathcal{V}}$ and $p_{\mathcal{V} \to \mathcal{V}_{\text{sub}}}$ be as above. For any tokens $y_{1:k} \in \mathcal{V}^*$, we have*

$$p_{\mathcal{V} \to \mathcal{V}_{\text{sub}}}(y_{1:k}*) = \sum_{x_{1:t} \in C_{\mathcal{V}, \mathcal{V}_{\text{sub}}}(y_{1:k})} p_{\mathcal{V}}(x_{1:t}*) \tag{16}$$

*Proof.* This is a generalization of Lemma 2.1. See the proof of Lemma A.1 □

The next problem is: how to efficiently compute the right hand side of Equation (16). To answer this, we introduce the following subsets of the relative covering $C_{\mathcal{V}, \mathcal{V}_{\text{sub}}}(y_{1:k})$:

$$C_{\mathcal{V}, \mathcal{V}_{\text{sub}}}^{\text{eq}}(y_{1:k}) := \{x_{1:t} \in C_{\mathcal{V}, \mathcal{V}_{\text{sub}}}(y_{1:k}) \mid [x_{1:t}]_{\mathcal{V} \to \mathcal{V}_{\text{sub}}} = y_{1:k}\}, \tag{17}$$

$$C_{\mathcal{V}, \mathcal{V}_{\text{sub}}}^{(l)}(y_{1:k}) := C_{\mathcal{V}, \mathcal{V}_{\text{sub}}}(y_{1:k}) \cap \mathcal{V}^l = \{x_{1:t} \in C_{\mathcal{V}, \mathcal{V}_{\text{sub}}}(y_{1:k}) \mid t = l\}. \tag{18}$$

Then the summands in Equation (16) can be decomposed into the following two cases.

**Lemma 3.3.** *$x_{1:t} \in C_{\mathcal{V}, \mathcal{V}_{\text{sub}}}(y_{1:k})$ if and only if $x_{1:t}$ satisfies either*

*(i) $x_{1:t} \in C_{\mathcal{V}, \mathcal{V}_{\text{sub}}}(y_{1:k-1})$ and $y_{1:k} \prec [x_{1:t}]_{\mathcal{V} \to \mathcal{V}_{\text{sub}}}$,*

*or (ii) $x_{1:t-1} \in C_{\mathcal{V}, \mathcal{V}_{\text{sub}}}^{\text{eq}}(y_{1:k-1})$ and $x_t \in C_{\mathcal{V}, \mathcal{V}_{\text{sub}}}^{(l=1)}(y_k)$.*

*Proof.* See the proof of Lemma A.2. □

Moreover, for the case (ii), we remark that the valid tokens $x_{1:t-1} \in C_{\mathcal{V}, \mathcal{V}_{\text{sub}}}^{\text{eq}}(y_{1:k-1})$ can be uniquely determined as $x_{1:t-1} = [[y_{1:k-1}]_{\mathcal{A}}]_{\mathcal{V}}$ for any valid[3] tokens $y_{1:k-1}$. Indeed, if $x_{1:t-1}$ satisfies $[x_{1:t-1}]_{\mathcal{V} \to \mathcal{V}_{\text{sub}}} = y_{1:k-1}$, we have $[x_{1:t-1}]_{\mathcal{A}} = [y_{1:k-1}]_{\mathcal{A}}$ and then $x_{1:t-1} = [[x_{1:t-1}]_{\mathcal{A}}]_{\mathcal{V}} = [[y_{1:k-1}]_{\mathcal{A}}]_{\mathcal{V}}$ by the validity of $x_{1:t-1}$. Conversely, if $x_{1:t-1} = [[y_{1:k-1}]_{\mathcal{A}}]_{\mathcal{V}} = y_{1:k-1}$ with the valid tokens $y_{1:k-1}$, we have $[x_{1:t-1}]_{\mathcal{V} \to \mathcal{V}_{\text{sub}}} = [[[y_{1:k-1}]_{\mathcal{A}}]_{\mathcal{V} \to \mathcal{V}_{\text{sub}}} = y_{1:k-1}$ by its validity.

By combining the above lemmas, we obtain the recursive formula to compute the desired probabilities:

---

[3]Here we consider the validity with respect to $\mathcal{T}_{\mathcal{V} \to \mathcal{V}_{\text{sub}}}$, i.e., $[[y_{1:k-1}]_{\mathcal{A}}]_{\mathcal{V} \to \mathcal{V}_{\text{sub}}} = y_{1:k-1}$.

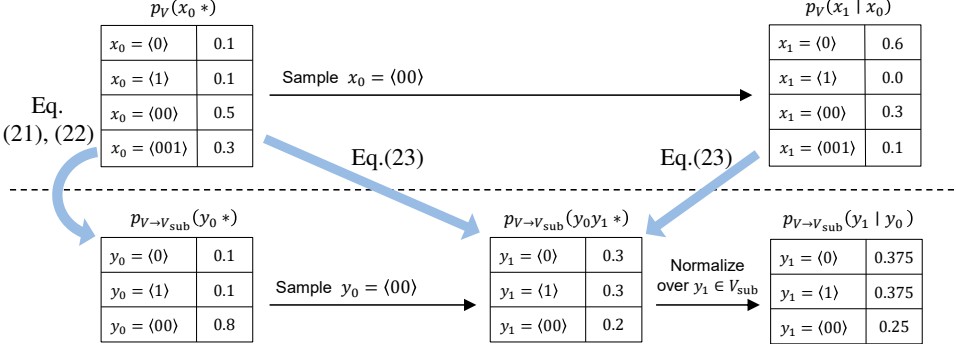

Figure 2: Summary of calculation for our example with $\mathcal{A} = \{0, 1\}$. The upper part shows the next-token distribution of the given language model $p_{\mathcal{V}}$ over $\mathcal{V} = \{\langle 0 \rangle, \langle 1 \rangle, \langle 00 \rangle, \langle 001 \rangle\}$, and the lower part shows the derived vocabulary-reduced model $p_{\mathcal{V} \to \mathcal{V}_{\mathrm{sub}}}$ over $\mathcal{V}_{\mathrm{sub}} = \{\langle 0 \rangle, \langle 1 \rangle, \langle 00 \rangle\}$.

**Theorem 3.4.** *For any tokens $y_{1:k} \in \mathcal{V}^*$ that is valid with respect to the nested tokenization $\mathcal{T}_{\mathcal{V} \to \mathcal{V}_{\mathrm{sub}}}$, we have*

$$p_{\mathcal{V} \to \mathcal{V}_{\mathrm{sub}}}(y_{1:k}*) = \sum_{\substack{x_{1:t} \in C_{\mathcal{V}, \mathcal{V}_{\mathrm{sub}}}(y_{1:k-1}) \\ \text{s.t. } y_{1:k} \prec [x_{1:t}]_{\mathcal{V} \to \mathcal{V}_{\mathrm{sub}}}}} p_{\mathcal{V}}(x_{1:t}*) + \sum_{\substack{x_t \in C^{(1)}_{\mathcal{V}, \mathcal{V}_{\mathrm{sub}}}(y_k), \\ \text{with } x_{1:t-1} := [[y_{1:k-1}]_{\mathcal{A}}]_{\mathcal{V}}}} p_{\mathcal{V}}(x_{1:t}*). \qquad (19)$$

*Proof.* By Lemma 3.2, the left-hand side is expanded as the sum of probabilities $p(x_{1:t}*)$ over the relative cover $x_{1:t} \in C_{\mathcal{V}, \mathcal{V}_{\mathrm{sub}}}(y_{1:k})$. Then we know that each summand $x_{1:t}$ satisfies either (i) or (ii) in Lemma 3.3, which leads to the decomposition shown in the right-hand side. □

**An illustrative example (Fig 2).** For simplicity, here we assume that $\mathcal{A} = \{0, 1\}$ instead of a set of bytes. Let $\mathcal{V} := \{\langle 0 \rangle, \langle 1 \rangle, \langle 00 \rangle, \langle 001 \rangle\}$ and $\mathcal{V}_{\mathrm{sub}} := \{\langle 0 \rangle, \langle 1 \rangle, \langle 00 \rangle\}$, where each $\langle - \rangle$ denotes a token. We suppose that the corresponding tokenizations are given by the greedy forward-matching tokenization, which maps each input bits $b_1 \cdots b_N \in \{0, 1\}^N$ to the longest matching tokens in the vocabulary, $\mathcal{V}$ or $\mathcal{V}_{\mathrm{sub}}$, greedily from left to right. Then we consider a language model $p_{\mathcal{V}}$ over $\mathcal{V}$ given by:

$$p_{\mathcal{V}}(x_0*) = \begin{cases} 0.1 & \text{if } x_0 = \langle 0 \rangle, \\ 0.1 & \text{if } x_0 = \langle 1 \rangle, \\ 0.5 & \text{if } x_0 = \langle 00 \rangle, \\ 0.3 & \text{if } x_0 = \langle 001 \rangle, \end{cases} \qquad p_{\mathcal{V}}(x_1 \mid x_0 = \langle 00 \rangle) = \begin{cases} 0.6 & \text{if } x_1 = \langle 0 \rangle, \\ 0 & \text{if } x_1 = \langle 1 \rangle, \\ 0.3 & \text{if } x_1 = \langle 00 \rangle, \\ 0.1 & \text{if } x_1 = \langle 001 \rangle, \end{cases} \qquad (20)$$

Note that the tokens $\langle 00 \rangle \langle 1 \rangle$ are invalid since $001$ is tokenized as $\langle 001 \rangle$ in the greedy forward-matching tokenization, and thus the probability $p_{\mathcal{V}}(x_1 = \langle 1 \rangle \mid x_0 = \langle 00 \rangle)$ is set to 0.

To compute $p_{\mathcal{V} \to \mathcal{V}_{\mathrm{sub}}}(y_0)$, we first calculate the relative covers:

$$C_{\mathcal{V}, \mathcal{V}_{\mathrm{sub}}}(\langle 0 \rangle) = \{\langle 0 \rangle\}, \ C_{\mathcal{V}, \mathcal{V}_{\mathrm{sub}}}(\langle 1 \rangle) = \{\langle 1 \rangle\}, \ C_{\mathcal{V}, \mathcal{V}_{\mathrm{sub}}}(\langle 00 \rangle) = \{\langle 00 \rangle, \langle 001 \rangle\}, \qquad (21)$$

Then we can compute the marginal probabilities $p_{\mathcal{V} \to \mathcal{V}_{\mathrm{sub}}}(y_0*)$ as

$$p_{\mathcal{V} \to \mathcal{V}_{\mathrm{sub}}}(y_0*) = \begin{cases} 0.1 \ (= p_{\mathcal{V}}(\langle 0 \rangle *)) & \text{if } y_0 = \langle 0 \rangle, \\ 0.1 \ (= p_{\mathcal{V}}(\langle 1 \rangle *)) & \text{if } y_0 = \langle 1 \rangle, \\ 0.8 \ (= p_{\mathcal{V}}(\langle 00 \rangle *) + p_{\mathcal{V}}(\langle 001 \rangle *)) & \text{if } y_0 = \langle 00 \rangle, \end{cases} \qquad (22)$$

Now suppose that $y_0 = \langle 00 \rangle$ is sampled. To derive the next-token probability $p_{\mathcal{V} \to \mathcal{V}_{\mathrm{sub}}}(y_1 \mid y_0 = \langle 00 \rangle)$, we need to consider the following relative covers:

$$C_{\mathcal{V}, \mathcal{V}_{\mathrm{sub}}}(\langle 00 \rangle \langle 0 \rangle) = \{\langle 00 \rangle \langle 0 \rangle\}, \ C_{\mathcal{V}, \mathcal{V}_{\mathrm{sub}}}(\langle 00 \rangle \langle 1 \rangle) = \{\langle 001 \rangle\},$$
$$C_{\mathcal{V}, \mathcal{V}_{\mathrm{sub}}}(\langle 00 \rangle \langle 00 \rangle) = \{\langle 00 \rangle \langle 00 \rangle, \langle 00 \rangle \langle 001 \rangle\},$$

**Algorithm 1** Lossless Vocabulary Reduction.

**Inputs:** Previously sampled tokens $y_{1:k} \in \mathcal{V}_{\mathrm{sub}}^k$, a global cache $\mathcal{P}$ for computed probabilities, a global cache $\mathcal{C}$ for relative minimal covers.
**Outputs:** $p_{\mathcal{V} \to \mathcal{V}_{\mathrm{sub}}}(y_{k+1} \mid y_{1:k})$ for $y_{k+1} \in \mathcal{V}_{\mathrm{sub}}$.
1: Tokenize $y_{1:k}$ in $\mathcal{V}$, i.e., $x_{1:t} := [[y_{1:k}]_{\mathcal{A}}]_{\mathcal{V}}$.
2: Compute $p_{\mathcal{V}}(x_{1:t}x*)$ for all $x \in \mathcal{V}$ by a single forward computation, and store them in $\mathcal{P}$.
3: Fetch $C_{\mathcal{V},\mathcal{V}_{\mathrm{sub}}}(y_{1:k})$ from the cache $\mathcal{C}$.
4: **for** $y_{k+1} \in \mathcal{V}_{\mathrm{sub}}$ **do**
5:     Initialize $\mathcal{S}[y_{k+1}] \leftarrow \emptyset$. ▷ *for collecting probs.*
6:     Initialize $C_{\mathcal{V},\mathcal{V}_{\mathrm{sub}}}(y_{1:k+1}) \leftarrow \emptyset$.
7:     **for** $x'_{1:t'} \in C_{\mathcal{V},\mathcal{V}_{\mathrm{sub}}}(y_{1:k})$ **do**
8:        **if** $y_{1:k+1} \prec [x'_{1:t'}]_{\mathcal{V} \to \mathcal{V}_{\mathrm{sub}}}$ **then**
9:           Fetch $p_{\mathcal{V}}(x'_{1:t'}*)$ from the cache $\mathcal{P}$.
10:           Add $x'_{1:t'}$ to $C_{\mathcal{V},\mathcal{V}_{\mathrm{sub}}}(y_{1:k+1})$.
11:           Add $p_{\mathcal{V}}(x'_{1:t'}*)$ to $\mathcal{S}[y_{k+1}]$.
12:     **for** $x_{t+1} \in \mathcal{V}$ **do**
13:        **if** $[x_{t+1}]_{\mathcal{V}_{\mathrm{sub}}} = y_{k+1} \cdots \in \mathcal{V}_{\mathrm{sub}}^*$ **then**
14:           Add $x_{1:t+1}$ to $C_{\mathcal{V},\mathcal{V}_{\mathrm{sub}}}(y_{1:k+1})$.
15:           Add $p_{\mathcal{V}}(x_{1:t}x_{t+1}*)$ to $\mathcal{S}[y_{k+1}]$.
16:     Store $C_{\mathcal{V},\mathcal{V}_{\mathrm{sub}}}(y_{1:k+1})$ in the cache $\mathcal{C}$.
17:     Set $\tilde{p}(y_{k+1}) \leftarrow \sum_{p \in \mathcal{S}[y_{k+1}]} p$. ▷ $p_{\mathcal{V},\mathcal{V}_{\mathrm{sub}}}(y_{1:k+1}*)$.
18: Set $p_{\mathcal{V},\mathcal{V}_{\mathrm{sub}}}(y_{k+1} \mid y_{1:k}) \leftarrow \tilde{p}(y_{k+1})/\sum_y \tilde{p}(y)$ for all $y_{k+1} \in \mathcal{V}_{\mathrm{sub}}$. ▷ *Marginalization.*
19: **return** $p_{\mathcal{V},\mathcal{V}_{\mathrm{sub}}}(\cdot \mid y_{1:k})$.

**Algorithm 2** Top-$K$ Approximation. ($K$-LVR)

**Inputs:** Same as Algorithm 1.
**Outputs:** Same as Algorithm 1.
1: Tokenize $y_{1:k}$ in $\mathcal{V}$, i.e., $x_{1:t} := [[y_{1:k}]_{\mathcal{A}}]_{\mathcal{V}}$.
2: Compute $p_{\mathcal{V}}(x_{1:t}x*)$ for all $x \in \mathcal{V}$ by a single forward computation, and store them in $\mathcal{P}$.
3: Fetch $C_{\mathcal{V},\mathcal{V}_{\mathrm{sub}}}(y_{1:k})$ from the cache $\mathcal{C}$.
4: **for** $y_{k+1} \in \mathcal{V}_{\mathrm{sub}}$ **do**
5:     Initialize $\mathcal{S}[y_{k+1}] \leftarrow \emptyset$. ▷ *for collecting probs.*
6:     Initialize $C_{\mathcal{V},\mathcal{V}_{\mathrm{sub}}}(y_{1:k+1}) \leftarrow \emptyset$.
7: **for** $x'_{1:t'} \in C_{\mathcal{V},\mathcal{V}_{\mathrm{sub}}}(y_{1:k})$ **do**
8:     $y_{k+1} \leftarrow$ the $(k+1)$-th sub-token of $x'_{1:t'}$
9:     Fetch $p_{\mathcal{V}}(x'_{1:t'}*)$ from the cache $\mathcal{P}$.
10:     Add $x'_{1:t'}$ to $C_{\mathcal{V},\mathcal{V}_{\mathrm{sub}}}(y_{1:k+1})$.
11:     Add $p_{\mathcal{V}}(x'_{1:t'}*)$ to $\mathcal{S}[y_{k+1}]$.
12: $\mathcal{V}^{(K)} \leftarrow \{x \in \mathcal{V} \mid p_{\mathcal{V}}(x_{1:t}x*) \text{ within the top-K.}\}$
13: **for** $x_{t+1} \in \mathcal{V}^{(K)}$ **do**
14:     $y_{k+1} \leftarrow$ the first sub-token of $[x_{t+1}]_{\mathcal{V}_{\mathrm{sub}}}$.
15:     Add $x_{1:t+1}$ to $C_{\mathcal{V},\mathcal{V}_{\mathrm{sub}}}(y_{1:k+1})$.
16:     Add $p_{\mathcal{V}}(x_{1:t}x_{t+1}*)$ to $\mathcal{S}[y_{k+1}]$.
17: **for** $y_{k+1} \in \mathcal{V}_{\mathrm{sub}}$ **do**
18:     Store $C_{\mathcal{V},\mathcal{V}_{\mathrm{sub}}}(y_{1:k+1})$ in the cache $\mathcal{C}$.
19:     Set $\tilde{p}(y_{k+1}) \leftarrow \sum_{p \in \mathcal{S}[y_{k+1}]} p$. ▷ $p_{\mathcal{V},\mathcal{V}_{\mathrm{sub}}}(y_{1:k+1}*)$.
20: Set $p_{\mathcal{V},\mathcal{V}_{\mathrm{sub}}}(y_{k+1} \mid y_{1:k}) \leftarrow \tilde{p}(y_{k+1})/\sum_y \tilde{p}(y)$ for all $y_{k+1} \in \mathcal{V}_{\mathrm{sub}}$. ▷ *Marginalization.*
21: **return** $p_{\mathcal{V},\mathcal{V}_{\mathrm{sub}}}(\cdot \mid y_{1:k})$.

Then the marginal probabilities $p_{\mathcal{V} \to \mathcal{V}_{\mathrm{sub}}}(\langle 00 \rangle y_1*)$ are obtained as follows:

$$p_{\mathcal{V} \to \mathcal{V}_{\mathrm{sub}}}(\langle 00 \rangle y_1*) = \begin{cases} 0.3 \,(= p_{\mathcal{V}}(\langle 00 \rangle \langle 0 \rangle *)) & \text{if } y_1 = \langle 0 \rangle, \\ 0.3 \,(= p_{\mathcal{V}}(\langle 001 \rangle *)) & \text{if } y_1 = \langle 1 \rangle, \\ 0.2 \,(= p_{\mathcal{V}}(\langle 00 \rangle \langle 00 \rangle *) + p_{\mathcal{V}}(\langle 00 \rangle \langle 001 \rangle *)) & \text{if } y_1 = \langle 00 \rangle, \end{cases} \qquad (23)$$

Finally, we obtain the next-token distribution by normalizing the above marginal probabilities:

$$p_{\mathcal{V} \to \mathcal{V}_{\mathrm{sub}}}(y_1 \mid y_0 = \langle 00 \rangle) = \begin{cases} 0.375 & \text{if } y_1 = \langle 0 \rangle, \\ 0.375 & \text{if } y_1 = \langle 1 \rangle, \\ 0.25 & \text{if } y_1 = \langle 00 \rangle, \end{cases} \qquad (24)$$

We can easily check that the probability of the output text starts from "000" is $0.5$ in both models $p_{\mathcal{V}}$ and $p_{\mathcal{V} \to \mathcal{V}_{\mathrm{sub}}}$, as expected by the lossless property (Theorem 3.1). For more details, see Appendix B.

### 3.3 Algorithm

Based on our theory of lossless vocabulary reduction given in the previous section, we derive an exact but computationally inefficient algorithm (Algorithm 1) and its efficient approximation (Algorithm 2) for practical autoregressive language models.

**Naive implementation.** Based on the theoretical results in previous sections, we can derive an algorithm (Algorithm 1) to recursively compute and sample from next-token distributions $p_{\mathcal{V} \to \mathcal{V}_{\mathrm{sub}}}$ over $\mathcal{V}_{\mathrm{sub}}$, given only access to some autoregressive language model with the vocabulary $\mathcal{V}$, i.e., next-token distributions $p_{\mathcal{V}}$ over $\mathcal{V}$. Due to its recursive nature, we can suppose that sub-tokens $y_{1:k} \in \mathcal{V}_{\mathrm{sub}}$ have been sampled in previous iterations by this algorithm, and its intermediate outcomes are properly cached. Under the circumstances, Algorithm 1 computes the next-token distribution $p_{\mathcal{V} \to \mathcal{V}_{\mathrm{sub}}}(y_{k+1} \mid y_{1:k})$ for all $y_{k+1} \in \mathcal{V}_{\mathrm{sub}}$ and sample a next token $y_{k+1}$ from it.

Now we explain how Algorithm 1 produces the desired next-token distribution. First of all, the given previous output $y_{1:k} \in \mathcal{V}_{\mathrm{sub}}$ is re-tokenized in $\mathcal{V}$, denoted by $x_{1:t} := [[y_{1:k}]_{\mathcal{A}}]_{\mathcal{V}}$, and then the next-token distribution $p_{\mathcal{V}}(x_{t+1} \mid x_{1:t})$ is computed by the given autoregressive language model (lines 1-2). Note that here is the only part that requires access to the given language model in Algorithm 1, and the results are cached in the global memory $\mathcal{P}$ so that they can be reused in future

iterations. Then, in lines 3-17, the marginal probability $p_{\mathcal{V} \to \mathcal{V}_{\mathrm{sub}}}(y_{1:k}y_{k+1}*)$ is computed for each $y_{k+1} \in \mathcal{V}_{\mathrm{sub}}$ by following Theorem 3.4. More precisely, lines 7-11 implement the first term in Equation (19), and lines 12-15 implement the second term. Finally, in line 18, the sum is normalized over $y_{k+1} \in \mathcal{V}_{\mathrm{sub}}$ to obtain the desired next-token distribution according to Equation (14).

**Efficient implementation.** Algorithm 1 has naively implemented the vocabulary reduction based on Equation (16). Although the implementation is straightforward from the theoretical perspective, it is computationally infeasible especially due to the two nested loops, lines 7-11 and lines 12-15. Here we discuss how to deform Algorithm 1 into a more efficiently approximated one, Algorithm 2.

First of all, we consider the first nested loop (lines 7-11 in Algorithm 1), which requires $|\mathcal{V}_{\mathrm{sub}}| \times |C_{\mathcal{V} \to \mathcal{V}_{\mathrm{sub}}}(y_{1:k})|$ iterations. This part can be separated from the outer loop over $y_{k+1} \in \mathcal{V}_{\mathrm{sub}}$, because the $y_{k+1}$ satisfying the condition in the line 8 is uniquely determined as the $(k+1)$-th sub-token of $[x'_{1:t'}]_{\mathcal{V} \to \mathcal{V}_{\mathrm{sub}}}$. Thus we can deform the lines 7-11 in Algorithm 1 into the lines 7-11 in Algorithm 2, which only requires $|C_{\mathcal{V} \to \mathcal{V}_{\mathrm{sub}}}(y_{1:k})|$ iterations.

Then we consider the second nested loop (lines 12-15 in Algorithm 1), which requires $|\mathcal{V}_{\mathrm{sub}}| \times |\mathcal{V}|$ iterations. This part can also be separated from the outer loop, because the $y_{k+1}$ satisfying the condition in the line 13 is uniquely determined as the first sub-token of $[x_{t+1}]_{\mathcal{V} \to \mathcal{V}_{\mathrm{sub}}}$. Thus the nested loop can be deformed into a single loop over $x_{t+1} \in \mathcal{V}$. Moreover, we observe that the token $x_{t+1}$ with a low probability can be ignored as it will not contribute to the final summation. Therefore we can restrict the single loop over $\mathcal{V}$ to the one over $\mathcal{V}^{(K)}$, the top-$K$ tokens with high probabilities. Based on these observations, we can deform the lines 12-15 in Algorithm 1 into the lines 12-16 in Algorithm 2, which only requires $K$ iterations. Here $K$ is an arbitrary hyperparameter and we will set it to be a small number $K = 300$ in our experiments. For ablation study with varying $K$, see Appendix D.7.

## 4 APPLICATION: ENSEMBLE VIA MAXIMAL COMMON VOCABULARY

Suppose that we have $N$ language models $p_{\mathcal{V}_1}, \dots, p_{\mathcal{V}_N}$, each defined over its own vocabulary $\mathcal{V}_i$. We assume that each $p_i$ satisfies the validity condition (7) with respect to a given tokenizer $\mathcal{T}_i = (\mathcal{V}_i, [-]_{\mathcal{V}_i}, [-]_{\mathcal{A}})$. For simplicity, we further assume that each tokenizer is given by Byte-Pair Encoding (BPE; Gage (1994); Sennrich et al. (2016)) with a set of token pairs $\mathcal{M}_i \subset \mathcal{V}_i^2$. Recall that, in BPE tokenization, a given text $a_1 \cdots a_n \in \mathcal{A}^n$ is first viewed as a sequence of byte tokens $x_1 \cdots x_n \in \mathcal{V}_i^n$, and then iteratively each adjacent tokens $x_t x_{t+1}$ satisfying $(x_t, x_{t+1}) \in \mathcal{M}_i$ are merged, i.e., replaced by the new token $x'_t := x_t x_{t+1} \in \mathcal{V}_i$.

To ensemble language models with different vocabularies, we introduce the *maximal common vocabulary* $\mathcal{V}_\cap$ and an associated tokenizer $\mathcal{T}_\cap = (\mathcal{V}_\cap, [-]_{\mathcal{V}_\cap}, [-]_{\mathcal{A}})$, so that it can be used in the nested tokenizer for lossless vocabulary reduction described in the previous section. Although there is some arbitrariness in the construction of such an associated tokenizer, we consider a BPE tokenizer constructed from the first tokenizer $\mathcal{T}_1$, with the vocabulary $\mathcal{V}_\cap$ and token pairs $\mathcal{M}_\cap$ given by:

$$\mathcal{V}_\cap := \bigcap_{i=1}^N \mathcal{V}_i, \quad \mathcal{M}_\cap := \{(y_1, y_2) \in \mathcal{M}_1 \mid y_1 y_2 \in \mathcal{V}_\cap\}. \tag{25}$$

Then the encoder $[-]_{\mathcal{V}_\cap}$ for $\mathcal{T}_\cap$ is defined by the iterative merging process of BPE, as explained above, with the token pairs $\mathcal{M}_\cap$. Although here we focused on a specific construction based on the BPE algorithm, the construction of $\mathcal{T}_\cap$ is not limited to BPE; other deterministic tokenizations such as the Unigram tokenization (Kudo, 2018) can also be employed with appropriate modifications.

Let $p_{\mathcal{V}_i \to \mathcal{V}_\cap}$ denote the vocabulary reduced model of $p_{\mathcal{V}_i}$ with the nested tokenizer $\mathcal{T}_{\mathcal{V} \to \mathcal{V}_\cap}$ with respect to $\mathcal{T}_\cap$, i.e., $[x_t]_{\mathcal{V}_i \to \mathcal{V}_\cap} := [[x_t]_{\mathcal{A}}]_{\mathcal{V}_\cap}$ for each token $x_t \in \mathcal{V}_i$. Each $p_{\mathcal{V}_i \to \mathcal{V}_\cap}$ is an independent language model over the common vocabulary $\mathcal{V}_\cap$ but exactly shares the quality of generated texts with the original model $p_{\mathcal{V}_i}$ by Theorem 3.1. Then we can define the ensemble[4] of the given $N$ language models $\{p_{\mathcal{V}_i}\}_{i=1,\cdots,N}$ as that of the vocabulary reduced models $\{p_{\mathcal{V}_i \to \mathcal{V}_\cap}\}_{i=1,\cdots,N}$:

$$p_{\mathrm{ens}}(y_{t+1} \mid y_{1:t}) \propto \prod_{i=1}^N p_{\mathcal{V}_i \to \mathcal{V}_\cap}(y_{t+1} \mid y_{1:t}), \quad \text{where } y_1, \cdots, y_{t+1} \in \mathcal{V}_\cap. \tag{26}$$

---

[4]In this paper, we mainly consider the ensemble by products of experts (Hinton, 1999), which is also known as logit-level ensemble for neural networks. See Appendix D.3 for more detailed discussion.

Figure 3: An example of text generation by greedy decoding of the original model (Llama3.2-3B) and the vocabulary-reduced models with varying maximal token lengths from 1 to 8 bytes. Each token is colored periodically for visibility. See Appendix D.4 for other examples.

Compared to the previous work of the byte-level ensemble (Phan et al., 2025; Vieira et al., 2025), our approach with the maximal common vocabulary enables faster generation because it can generate multiple bytes by a single inference of the ensemble model, with almost the same inference cost of each vocabulary reduced model as the corresponding byte-level model.

## 5 EXPERIMENTS

In this section, we conduct experiments to validate (i) whether our proposed method can reduce given vocabularies to various sub-vocabularies without loss in accuracy and (ii) whether our proposed method can be effectively applied to token-level ensemble. See Appendix C for experimental details.

### 5.1 EXPERIMENTS ON VOCABULARY REDUCTION

Given a language model with a vocabulary $\mathcal{V}$, we consider the $N$-bytes sub-vocabulary $\mathcal{V}_{\leq N} := \{v \in \mathcal{V} \mid \text{len}([v]_{\mathcal{A}}) \leq N\}^5$ and assess the capability of vocabulary reduction from $\mathcal{V}$ to $\mathcal{V}_{\leq N}$ with $N = 1, 2, 4, 8$. As a baseline for vocabulary reduction, we consider **Naive Restriction** that simply puts zeros to the probabilities of excluded tokens $\mathcal{V} \setminus \mathcal{V}_{\leq N}$ and renormalizes the remaining probabilities over $\mathcal{V}_{\leq N}$. In Table 1, we evaluated the accuracy of vocabulary-reduced models by the above baseline (Naive) and our algorithm ($K$-LVR) on the benchmark GSM8K. The results show that our algorithm ($K$-LVR) overall achieves almost same accuracy as the original model over the full vocabulary. In Figure 3, we show an example of greedy decoding for $K$-LVR models with the sub-vocabularies $\mathcal{V}_{\leq N}$, and observe the consistent results except for the 1-byte case. In the 1-byte case, the generated text deviates from the original one after the first sentence due to the nature of greedy decoding[6], but the final answer is still consistent with the original one as Theorem 3.1 implied.

### 5.2 EXPERIMENTS ON ENSEMBLE VIA MAXIMAL COMMON VOCABULARY (MCV)

In Table 2, as an application of the vocabulary reduction, we conducted experiments with ensemble of two language models of similar accuracy: Qwen2.5-3B and Falcon3-7B. The former model has 151,665 tokens and the latter has 131,072 tokens, whose maximal common vocabulary consists of 63,552 tokens. **Union** and **Naive (MCV)** refer to the heuristic baselines described in Appendix C.2. From the results, we can see that (i) ensemble with $K$-LVR over MCV overall achieves comparable accuracy to the byte-level ensemble (Phan et al., 2025) and other baselines, and (ii) the heuristic baselines catastrophically fail in some cases, while our $K$-LVR consistently works well even in such cases. Moreover, as shown in Table 3 in Appendix, we observed that the ensemble over MCV is actually faster than the byte-level ensemble as we expected at first, which highlights the benefit of generalization to arbitrary sub-vocabularies beyond one-bytes.

## 6 RELATED WORK

**Vocabulary reduction.** There has been a line of research on vocabulary reduction, specifically aiming for compressing the size of language models, for e.g., Ataman et al. (2017); Gee et al.

---

[5]Here we define $\text{len}(a_1 \cdots a_N) := N$ for $a_1 \cdots a_N \in \mathcal{A}^*$.

[6]See Appendix B for detailed discussion on the inconsistency occured in greedy decoding.

| Models | Methods | Full | 1 Bytes | ≤ 2 Bytes | ≤ 4 Bytes | ≤ 8 Bytes |
|---|---|---|---|---|---|---|
| OLMo2-1B | Naive | 30.40 | 0.00 | 0.00 | 7.28 | 30.55 |
| | $K$-LVR | | **30.40** | **31.46** | **31.39** | **31.77** |
| Llama3.2-3B | Naive | 26.00 | 0.00 | 0.00 | 11.52 | **26.31** |
| | $K$-LVR | | **26.31** | **26.23** | **25.93** | 26.16 |
| Qwen2.5-3B | Naive | 71.27 | 0.00 | 0.00 | 23.65 | 65.96 |
| | $K$-LVR | | **71.19** | **70.43** | **71.42** | **72.18** |
| Falcon3-7B | Naive | 76.65 | 0.00 | 0.00 | 34.72 | 72.40 |
| | $K$-LVR | | **78.92** | **79.38** | **79.30** | **79.23** |

Table 1: Results of vocabulary reduction on GSM8K, with varying maximal token lengths from 1 to 8 bytes. **Full** refers to the original models, **Naive** is the baseline of naive restriction and $K$**-LVR** is our algorithm.

| Single | GSM8K | MATH | ACP | MMLU-Pro |
|---|---|---|---|---|
| Qwen2.5-3B | 71.27 | **27.86** | **36.71** | 37.71 |
| Falcon3-7B | **76.65** | 26.14 | 36.29 | **42.71** |
| *Ensemble (PoE)* | | | | |
| Union (Yao et al., 2025) | 27.60 | 19.57 | 25.86 | 1.93 |
| Naive (MCV) | 24.94 | 21.71 | 25.71 | 2.07 |
| $K$-LVR (1-Bytes) | **82.49** | **30.71** | **35.43** | 41.21 |
| $K$-LVR (MCV) | 81.12 | 30.29 | 34.71 | **42.00** |

Table 2: Results of ensemble by product of experts (PoE; Hinton (1999)). **Union** refers to the baseline of the union vocabulary and **Naive (MCV)** refers to the baseline of naive restriction to the MCV. (Appendix C.2)

(2022); Ushio et al. (2023); Bogoychev et al. (2024); Chizhov et al. (2024); Nozaki et al. (2025), while their motivation differs from ours. Some of them proposed methods like naive restriction with heuristics of vocabulary selection (Ushio et al., 2023; Bogoychev et al., 2024), and others proposed to modify embedding vectors with additional training to keep accuracy. Recently, apart from this line of research, several work (Phan et al., 2025; Vieira et al., 2025; Hayase et al., 2025) have proposed methods to convert token-level language models into the equivalent byte-level language models in inference-time. Phan et al. (2025) derived the first efficient method for the byte-level reduction under the validity assumption, and empirically confirmed its applicability to ensemble with different vocabularies. Vieira et al. (2025) derived the byte-level reduction under general condition, while it requires more computation for accurate conversion. Our framework can be seen as a generalization of Phan et al. (2025) from bytes to arbitrary sub-vocabularies under the validity assumption.

**Ensemble with different vocabularies.** Specifically for ensemble of language models with different vocabularies, previous work have taken several heuristic approaches based on (i) partial matching between tokens as strings (Jiang et al., 2023; Wan et al., 2024; Liu et al., 2025), (ii) similarity between tokens in the shared embedding space (Xu et al., 2024; Huang et al., 2024), and (iii) the union vocabulary (Yu et al., 2024; Yao et al., 2025). Although the first and second approaches seem to work well in experiments, they have no theoretical guarantee for their success and it is also difficult for them to be applied beyond ensemble. The second approach also requires models to share the same embedding space. The third approach, which simply extends next-token distributions by putting zero probabilityies to out-of-vocabulary tokens, has been reported to beat the previous heuristic methods despite of its simplicity (Yao et al., 2025). However, by its nature, it possibly struggles to capture the mutual relations between out-of-vocabulary tokens since it completely ignores any partial match as strings, which may lead to the perfomance drop observed in our experiments.

# 7 CONCLUSION

In this paper, we established the first theoretical framework of lossless vocabulary reduction, which reduces a given next-token distribution to the one with an arbitrary sub-vocabulary while preserving the generation quality. Compared to the previous byte-level reduction, our framework enables more flexible and efficient cooperation between different language models through their common vocabularies. We hope that our work will open up a new research direction for efficient lossless cooperation between language models with different vocabularies in a principled way.

## REPRODUCIBILITY STATEMENT

The experimental details are provided in Appendix C. All proofs are provided in Sections 3 and A.

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

## A    PROOFS FOR SECTION 3

**Lemma A.1.** *Let $p_\mathcal{V}$ and $p_{\mathcal{V}\to\mathcal{V}_{\mathrm{sub}}}$ be as in Lemma 3.2. For any tokens $y_{1:k} \in \mathcal{V}^*$, we have*

$$p_{\mathcal{V}\to\mathcal{V}_{\mathrm{sub}}}(y_{1:k}*) = \sum_{x_{1:t} \in C_{\mathcal{V},\mathcal{V}_{\mathrm{sub}}}(y_{1:k})} p_\mathcal{V}(x_{1:t}*). \tag{27}$$

*Proof.* Here we introduce the following notation:

$$C^{(t)}_{\mathcal{V},\mathcal{V}_{\mathrm{sub}}}(y_{1:k}) := C_{\mathcal{V},\mathcal{V}_{\mathrm{sub}}}(y_{1:k}) \cap \mathcal{V}^t, \tag{28}$$

Then we have the following decomposition for the relative covering set:

$$C_{\mathcal{V},\mathcal{V}_{\mathrm{sub}}}(y_{1:k}) = \bigsqcup_{t\in\mathbb{N}} C^{(t)}_{\mathcal{V},\mathcal{V}_{\mathrm{sub}}}(y_{1:k})$$

Using these notation, we have

$$
\begin{aligned}
p_{\mathcal{V}\to\mathcal{V}_{\mathrm{sub}}}(y_{1:k}*) &= \sum_{y_{1:K}\in y_{1:k}*} p_{\mathcal{V}\to\mathcal{V}_{\mathrm{sub}}}(y_{1:K}) \\
&= \sum_{\substack{y_{1:K}\in y_{1:k}* \\ x_{1:T}\in\mathcal{V}^*, \\ [x_{1:T}]_{\mathcal{V}\to\mathcal{V}_{\mathrm{sub}}}=y_{1:K}}} p_\mathcal{V}(x_{1:T}) \\
&= \sum_{\substack{x_{1:T}\in\mathcal{V}^*, \\ [x_{1:T}]_{\mathcal{V}\to\mathcal{V}_{\mathrm{sub}}}\in y_{1:k}*}} p_\mathcal{V}(x_{1:T}) \\
&= \sum_{\substack{x_{1:T}\in\mathcal{V}^*, \\ [x_1]_{\mathcal{V}\to\mathcal{V}_{\mathrm{sub}}}\cdots[x_T]_{\mathcal{V}\to\mathcal{V}_{\mathrm{sub}}}\in y_{1:k}*}} p_\mathcal{V}(x_{1:T}) \\
&= \sum_{t\in\mathbb{N}} \sum_{\substack{x_{1:T}\in\mathcal{V}^*, \\ [x_1]_{\mathcal{V}\to\mathcal{V}_{\mathrm{sub}}}\cdots[x_t]_{\mathcal{V}\to\mathcal{V}_{\mathrm{sub}}}\in y_{1:k}*, \\ [x_1]_{\mathcal{V}\to\mathcal{V}_{\mathrm{sub}}}\cdots[x_t-1]_{\mathcal{V}\to\mathcal{V}_{\mathrm{sub}}}\notin y_{1:k}*}} p_\mathcal{V}(x_{1:T}) \\
&= \sum_{t\in\mathbb{N}} \sum_{\substack{x_{1:t}\in\mathcal{V}^t, \\ [x_1]_{\mathcal{V}\to\mathcal{V}_{\mathrm{sub}}}\cdots[x_t]_{\mathcal{V}\to\mathcal{V}_{\mathrm{sub}}}\in y_{1:k}*, \\ [x_1]_{\mathcal{V}\to\mathcal{V}_{\mathrm{sub}}}\cdots[x_t-1]_{\mathcal{V}\to\mathcal{V}_{\mathrm{sub}}}\notin y_{1:k}*}} \sum_{x_{t+1:T}\in\mathcal{V}^*} p_\mathcal{V}(x_{1:T}) \\
&= \sum_{t\in\mathbb{N}} \sum_{x_{1:t}\in C^{(t)}_{\mathcal{V},\mathcal{V}_{\mathrm{sub}}}(y_{1:k})} p_\mathcal{V}(x_{1:t}*) \\
&= \sum_{x_{1:t}\in C_{\mathcal{V},\mathcal{V}_{\mathrm{sub}}}(y_{1:k})} p_\mathcal{V}(x_{1:t}*)
\end{aligned}
$$

$\square$

**Lemma A.2.** *Under the notation in Section 3, $x_{1:t} \in C_{\mathcal{V},\mathcal{V}_{\mathrm{sub}}}(y_{1:k})$ if and only if $x_{1:t}$ satisfies either*

*(i)* $x_{1:t} \in C_{\mathcal{V},\mathcal{V}_{\mathrm{sub}}}(y_{1:k-1})$ *and* $y_{1:k} \prec [x_{1:t}]_{\mathcal{V}\to\mathcal{V}_{\mathrm{sub}}}$,

*or (ii)* $x_{1:t}$ *is valid and satisfies* $[x_{1:t-1}]_{\mathcal{V}\to\mathcal{V}_{\mathrm{sub}}} = y_{1:k-1}$ *and* $y_k \prec [x_t]_{\mathcal{V}_{\mathrm{sub}}}$.

*Proof.* First of all, we note that $y_{1:k'} = [x_{1:t-1}]_{\mathcal{V}\to\mathcal{V}_{\mathrm{sub}}}$ for some $k' < k$ by the assumption $x_{1:t} \in C_{\mathcal{V},\mathcal{V}_{\mathrm{sub}}}(y_{1:k})$. Then the last token $x_t$ can be expanded as $[x_t]_{\mathcal{V}\to\mathcal{V}_{\mathrm{sub}}} = y_{k'+1:k} \cdots \in y_{k'+1:k}*$. In the case of $k' = k - 1$, it follows that $y_k \prec [x_t]_{\mathcal{V}\to\mathcal{V}_{\mathrm{sub}}}$ and thus corresponds to the case (ii). On the other hand, if $k' < k - 1$, we have $x_{1:t} \in C_{\mathcal{V},\mathcal{V}_{\mathrm{sub}}}(y_{1:k-1})$ because $[x_{1:t}]_{\mathcal{V}\to\mathcal{V}_{\mathrm{sub}}} \in y_{1:k-1}*$ and $[x_{1:t-1}]_{\mathcal{V}\to\mathcal{V}_{\mathrm{sub}}} = y_{1:k'} \notin y_{1:k-1}*$, which corresponds to the case (i). $\square$

## B  AN EXAMPLE OF LOSSLESS VOCABULARY REDUCTION

Here we assume that $\mathcal{A} = \{0, 1\}$ instead of a set of bytes. Let $\mathcal{V} := \{\langle 0 \rangle, \langle 1 \rangle, \langle 00 \rangle, \langle 001 \rangle\}$ and $\mathcal{V}_{\mathrm{sub}} := \{\langle 0 \rangle, \langle 1 \rangle, \langle 00 \rangle\}$, where each $\langle - \rangle$ denotes a token. We suppose that the corresponding tokenizations are given by the greedy forward-matching tokenization, which maps each input bits $b_1 \cdots b_N \in \{0, 1\}^N$ to the longest matching tokens in the vocabulary, $\mathcal{V}$ or $\mathcal{V}_{\mathrm{sub}}$, from left to right. Then we consider a language model $p_\mathcal{V}$ over $\mathcal{V}$ given by:

$$p_\mathcal{V}(x_0 *) = \begin{cases} 0.1 & \text{if } x_0 = \langle 0 \rangle, \\ 0.1 & \text{if } x_0 = \langle 1 \rangle, \\ 0.5 & \text{if } x_0 = \langle 00 \rangle, \\ 0.3 & \text{if } x_0 = \langle 001 \rangle, \end{cases} \qquad p_\mathcal{V}(x_1 \mid x_0 = \langle 00 \rangle) = \begin{cases} 0.6 & \text{if } x_1 = \langle 0 \rangle, \\ 0 & \text{if } x_1 = \langle 1 \rangle, \\ 0.3 & \text{if } x_1 = \langle 00 \rangle, \\ 0.1 & \text{if } x_1 = \langle 001 \rangle, \end{cases} \tag{29}$$

Note that the tokens $\langle 00 \rangle \langle 1 \rangle$ are invalid since 001 is tokenized as $\langle 001 \rangle$ in the greedy forward-matching tokenization, and thus the probability $p_\mathcal{V}(x_1 = \langle 1 \rangle \mid x_0 = \langle 00 \rangle)$ is set to 0.

To compute $p_{\mathcal{V} \rightarrow \mathcal{V}_{\mathrm{sub}}}(y_0)$, we first calculate the relative covers:

$$C_{\mathcal{V}, \mathcal{V}_{\mathrm{sub}}}(\langle 0 \rangle) = \{\langle 0 \rangle\}, \; C_{\mathcal{V}, \mathcal{V}_{\mathrm{sub}}}(\langle 1 \rangle) = \{\langle 1 \rangle\}, \; C_{\mathcal{V}, \mathcal{V}_{\mathrm{sub}}}(\langle 00 \rangle) = \{\langle 00 \rangle, \langle 001 \rangle\}, \tag{30}$$

Then we can compute the marginal probabilities $p_{\mathcal{V} \rightarrow \mathcal{V}_{\mathrm{sub}}}(y_0 *)$ as

$$p_{\mathcal{V} \rightarrow \mathcal{V}_{\mathrm{sub}}}(y_0 *) = \begin{cases} 0.1 \; (= p_\mathcal{V}(\langle 0 \rangle *)) & \text{if } y_0 = \langle 0 \rangle, \\ 0.1 \; (= p_\mathcal{V}(\langle 1 \rangle *)) & \text{if } y_0 = \langle 1 \rangle, \\ 0.8 \; (= p_\mathcal{V}(\langle 00 \rangle *) + p_\mathcal{V}(\langle 001 \rangle *)) & \text{if } y_0 = \langle 00 \rangle, \end{cases} \tag{31}$$

Now suppose that $y_0 = \langle 00 \rangle$ is sampled. To derive the next-token probability $p_{\mathcal{V} \rightarrow \mathcal{V}_{\mathrm{sub}}}(y_1 \mid y_0 = \langle 00 \rangle)$, we need to consider the following relative covers:

$$C_{\mathcal{V}, \mathcal{V}_{\mathrm{sub}}}(\langle 00 \rangle \langle 0 \rangle) = \{\langle 00 \rangle \langle 0 \rangle\}, \; C_{\mathcal{V}, \mathcal{V}_{\mathrm{sub}}}(\langle 00 \rangle \langle 1 \rangle) = \{\langle 001 \rangle\},$$
$$C_{\mathcal{V}, \mathcal{V}_{\mathrm{sub}}}(\langle 00 \rangle \langle 00 \rangle) = \{\langle 00 \rangle \langle 00 \rangle, \langle 00 \rangle \langle 001 \rangle\},$$

Then the marginal probabilities $p_{\mathcal{V} \rightarrow \mathcal{V}_{\mathrm{sub}}}(\langle 00 \rangle y_1 *)$ is obtained as follows:

$$p_{\mathcal{V} \rightarrow \mathcal{V}_{\mathrm{sub}}}(\langle 00 \rangle y_1 *) = \begin{cases} 0.3 \; (= p_\mathcal{V}(\langle 00 \rangle \langle 0 \rangle *)) & \text{if } y_1 = \langle 0 \rangle, \\ 0.3 \; (= p_\mathcal{V}(\langle 001 \rangle *)) & \text{if } y_1 = \langle 1 \rangle, \\ 0.2 \; (= p_\mathcal{V}(\langle 00 \rangle \langle 00 \rangle *) + p_\mathcal{V}(\langle 00 \rangle \langle 001 \rangle *)) & \text{if } y_1 = \langle 00 \rangle, \end{cases} \tag{32}$$

Finally, we obtain the next-token distribution by normalizing the above marginal probabilities:

$$p_{\mathcal{V} \rightarrow \mathcal{V}_{\mathrm{sub}}}(y_1 \mid y_0 = \langle 00 \rangle) = \begin{cases} 0.375 & \text{if } y_1 = \langle 0 \rangle, \\ 0.375 & \text{if } y_1 = \langle 1 \rangle, \\ 0.25 & \text{if } y_1 = \langle 00 \rangle, \end{cases} \tag{33}$$

Now we compare the corresponding text distributions for $p_\mathcal{V}$ and $p_{\mathcal{V} \rightarrow \mathcal{V}_{\mathrm{sub}}}$. Especially, we calculate the probability that the output text starts with "000". For the case of $p_\mathcal{V}$, such a probability is nothing but the probability of $x_0 = \langle 00 \rangle$, which is 0.5, since the text "$000 \cdots$" can be tokenized as either $\langle 00 \rangle \langle 0 \rangle \cdots$ or $\langle 00 \rangle \langle 00 \rangle$ or $\langle 00 \rangle \langle 001 \rangle \cdots$, by the greedy forward-matching tokenization. On the other hand, for the case of $p_{\mathcal{V} \rightarrow \mathcal{V}_{\mathrm{sub}}}$, the text "$000 \cdots$" can be tokenized as either $\langle 00 \rangle \langle 0 \rangle \cdots$ or $\langle 00 \rangle \langle 00 \rangle \cdots$, and thus the corresponding probability is given by $p_{\mathcal{V} \rightarrow \mathcal{V}_{\mathrm{sub}}}(\langle 00 \rangle \langle 0 \rangle *) + p_{\mathcal{V} \rightarrow \mathcal{V}_{\mathrm{sub}}}(\langle 00 \rangle \langle 00 \rangle *) = 0.3 + 0.2 = 0.5$. Therefore, the two distributions $p_\mathcal{V}$ and $p_{\mathcal{V} \rightarrow \mathcal{V}_{\mathrm{sub}}}$ are equally plausible to output the text starting with "000".

Also, here we can observe the following fact: even if two token-distributions are equivalent at the level of their byte-level distributions, *the texts obtained by greedy decoding may be different from each other in general*. Indeed, on the one hand, the greedy decoding for the original distribution $p_\mathcal{V}$ generates the first token $x_0 = \langle 00 \rangle$ and the second token $x_1 = \langle 0 \rangle$ according to equation (29). On the other hand, the greedy decoding for the reduced distribution $p_{\mathcal{V} \rightarrow \mathcal{V}_{\mathrm{sub}}}$ may generate[7] $y_0 = \langle 00 \rangle$ and $y_1 = \langle 1 \rangle$ according to equation (31) and (32), resulting in a different text 001 from the former one 000, even though both distributions have the same text distribution by Theorem 3.1.

---

[7]This behavior is dependent on the implementation of greedy decoding sicne there are two tokens ($y_1 = \langle 0 \rangle$ and $y_1 = \langle 0 \rangle$) achieving the maximal conditional probability.

## C EXPERIMENTAL DETAILS

### C.1 SETUPS

**Models.**

- **Qwen 2.5** (Yang et al., 2024): A family of large language models developed by Qwen Team, ranging from 0.5B to 72B parameters. The tokenizer is implemented by the byte-level BPE with the vocabulary consisting of 151,665 tokens.

- **OLMo 2** (OLMo et al., 2024): A family of large language models developed by Allen Institute for AI, ranging from 1B to 32B parameters. The tokenizer is implemented by the byte-level BPE with the vocabulary consisting of 100,278 tokens.

- **Llama 3.1 and 3.2** (Meta, 2024a;b): A family of large language models developed by Meta AI, ranging from 1B to 90B parameters. The tokenizer is implemented by the byte-level BPE with the vocabulary consisting of 128,256 tokens.

- **Falcon 3** (Falcon-LLM, 2024): A family of large language models developed by Technology Innovation Institute (TII), ranging from 1B to 10B parameters. The tokenizer is implemented by the byte-level BPE with the vocabulary consisting of 131,072 tokens.

- **Phi 2** (Javaheripi et al., 2023): A large language model with 3B parameters, developed by Microsoft. The tokenizer is implemented by the byte-level BPE with the vocabulary consisting of 50,295 tokens.

- **Yi 1.5** (Young et al., 2024): A family of large language models developed by 01.AI, ranging from 6B to 34B parameters. The tokenizer is implemented by the byte-level BPE with the vocabulary consisting of 63,992 tokens.

**Datasets.** We used the following datasets for evaluating language models through the lm-evaluation-harness library (Gao et al., 2024) with default options.

- **GSM8K** (Cobbe et al., 2021): A dataset consisting of grade-school math questions. Each question has an example of chain-of-thought argument followed by an open-ended answer. For each question, 5 pairs of a question and its answer are provided for language models by default. We reported the percentage of strictly-extracted correct answers.

- **MATH** (Hendrycks et al., 2021b): A dataset of mathematical problem-solving tasks in 7 categories that require numerical and logical reasoning skills. Each question has an example of chain-of-thought argument followed by an open-ended answer. For each question, 4 pairs of a question and its answer are provided for language models, following the setting by Lewkowycz et al. (2022). We reported the percentage of correct answers after which processed symbolically, with randomly sampled 100 questions from each category.

- **ACPBench** (Kokel et al., 2025): A dataset of tasks in 7 categories, evaluating the reasoning ability about action, change, and planning, with multiple answer choices. Each question has an example of chain-of-thought argument followed by the correct answer. For each question, 2 pairs of a question and its answer are provided for language models by default. We reported the percentage of correct answers for the questions with multiple choices, with randomly sampled 100 questions from each category.

- **MMLU-Pro** (Wang et al., 2024): A dataset of question-answering tasks in 14 categories, enhancing MMLU (Hendrycks et al., 2021a) by more challenging questions that require reasoning skills with multiple answer choices. Each question has an example of chain-of-thought argument followed by the correct answer. For each question, 5 pairs of a question and its answer are provided for language models by default. We reported the percentage of correct answers, with randomly sampled 100 questions from each category.

**Decoding method.** Unless otherwise stated, we employed the greedy decoding for sampling tokens, i.e., iteratively sampling tokens with the highest probability, for both simpliticty and reproducibility. It is noteworthy that, even if two token distributions have the same text distribution, the resulting texts by greedy decoding for these models may be different from each other, as discussed in Appendix B.

## C.2 BASELINES

**Baselines for vocabulary reduction.**

- **Naive Restriction**: Naive restriction is the most straightforward method for vocabulary reduction, but with no theoretical guarantee for accuracy. Let $p_{\mathcal{V}}(x_{t+1}|x_1 \cdots x_t)$ be a next-token distribution over $\mathcal{V}$, and let $\mathcal{V}_{\text{sub}}$ be a subset of $\mathcal{V}$. The naive restriction $p_{\mathcal{V}_{\text{sub}}}(y_{k+1}|y_1 \cdots y_k)$ is defined as follows: Given previously sampled tokens $y_1 \cdots y_k$, we first retokenize it in $\mathcal{V}$ by $x_1 \cdots x_t := [[y_1 \cdots y_k]_{\mathcal{A}}]_{\mathcal{V}}$. Then we compute the next-token distribution $p_{\mathcal{V}}(x_{t+1}|x_1 \cdots x_t)$, forcibly replace $p_{\mathcal{V}}(x_{t+1}|x_1 \cdots x_t)$ by zero for all $x_{t+1} \notin \mathcal{V}_{\text{sub}}$, and renormalize it. Finally, we sample a next token $y_{k+1} = x_{t+1} \in \mathcal{V}_{\text{sub}}$ from the renormalized distribution.

- **Byte-Level Reduction** (Phan et al., 2025): Since our lossless vocabulary reduction (LVR) can be seen as a generalization of the byte-level reduction proposed in Phan et al. (2025), LVR with 1-byte tokens is actually equivalent to the byte-level reduction.

**Baselines for ensemble with different vocabularies.**

- **Ensemble over Union Vocabulary** (Yu et al., 2024; Yao et al., 2025): Given language models $p_{\mathcal{V}_i}$ with different vocabularies $\mathcal{V}_i$, we can simply extend each next-token distribution $p_{\mathcal{V}_i}(x_{t+1}|x_1 \cdots x_t)$ to the one over the union vocabulary $\mathcal{V}_{\cup} := \bigcup_i \mathcal{V}_i$ by putting zero probabilities for $x_{t+1} \in \mathcal{V}_{\cup} \setminus \mathcal{V}_i$. More specifically, to extend next-token distributions of the $i$-th model, given previous samples $y_1 \cdots y_k \in \mathcal{V}_{\cup}^*$, we first take the retokenization $x_1 \cdots x_t := [[y_1 \cdots y_k]_{\mathcal{A}}]_{\mathcal{V}_i}$ and then compute the next-token distribution of the $i$-th model, following the procedure in Yao et al. (2025).

- **Ensemble with Naive Restriction over MCV**: We just apply the naive restriction with $\mathcal{V}_{\text{sub}} := \mathcal{V}_{\cap}$ (as defined in Section 4) to each language model, and then compute their ensemble distribution.

# D  ADDITIONAL EXPERIMENTAL RESULTS

## D.1  INFERENCE EFFICIENCY

|  | OLMo2-1B & Qwen2.5-0.5B | Qwen2.5-3B & Falcon3-7B |
|---|---|---|
| Ensemble (1-Byte) | $25.72 \pm 1.78$ (bytes/sec) | $17.03 \pm 0.70$ (bytes/sec) |
| Ensemble (MCV) | $46.78 \pm 9.29$ (bytes/sec) | $33.50 \pm 5.41$ (bytes/sec) |

Table 3: Inference speed over 100 questions on GSM8K. Since the average bytes of tokens in maximal common vocabulary (MCV) is obviously greater than 1 byte, the ensemble over MCV can generate more bytes per second than the byte-level ensemble (Phan et al., 2025).

## D.2  EXTENDED RESULTS ON VOCABULARY REDUCTION

| Models | Methods | Full | 1 Bytes | $\leq$ 2 Bytes | $\leq$ 4 Bytes | $\leq$ 8 Bytes |
|---|---|---|---|---|---|---|
| Qwen2.5-0.5B | Naive | 34.27 | 0.00 | 0.00 | 6.90 | 30.86 |
|  | $K$-LVR |  | **34.95** | **33.43** | **34.34** | **34.34** |
| OLMo2-1B | Naive | 30.40 | 0.00 | 0.00 | 7.28 | 30.55 |
|  | $K$-LVR |  | **30.40** | **31.46** | **31.39** | **31.77** |
| Llama3.2-3B | Naive | 26.00 | 0.00 | 0.00 | 11.52 | **26.31** |
|  | $K$-LVR |  | **26.31** | **26.23** | **25.93** | 26.16 |
| Qwen2.5-3B | Naive | 71.27 | 0.00 | 0.00 | 23.65 | 65.96 |
|  | $K$-LVR |  | **71.19** | **70.43** | **71.42** | **72.18** |
| Falcon3-7B | Naive | 76.65 | 0.00 | 0.00 | 34.72 | 72.40 |
|  | $K$-LVR |  | **78.92** | **79.38** | **79.30** | **79.23** |

Table 4: Quantitative evaluations of vocabulary reduction on GSM8K, with varying maximal token lengths from 1 to 8 bytes. **Full** refers to the original models, **Naive** is the baseline of naive restriction (Appendix C) and $K$-**LVR** is our algorithm.

## D.3  PRODUCTS OF EXPERTS AND MIXTURES OF EXPERTS

There are two straightforward definitions for ensemble of probability distributions $p_1(x), \cdots, p_n(x)$ over the same domain $X$:

- Products of Experts (PoE; Hinton (1999)) with uniform weights:

$$p_{\text{ens}}(x) \propto \prod_{i=1}^{n} p_i(x)$$

- Mixtures of Experts (MoE; Jordan & Jacobs (1994)) with uniform weights:

$$p_{\text{ens}}(x) \propto \sum_{i=1}^{n} p_i(x)$$

Intuitively, PoE corresponds to the probability of sampling the same $x$ simultaneously from all distributions $p_i(x)$, and MoE corresponds to the probability of sampling $x$ from some randomly-chosen distribution $p_i(x)$. As argued in Hinton (1999), the former ensemble is more suitable to high-dimensional probabilities, including next-token distributions over the vocabulary of more than hundreds or thousands tokens, than the latter ensemble. Indeed, in the case of next-token distributions, a sampled token from the MoE model is preferred by some next-token distribution $p_{i_0}(x)$ but may be unfavored by the other distributions, which causes the distributional shift in the next sampling phase.

In addition to the results in the main paper, we performed experiments of both PoE and MoE with small models (Table 5) and larger models (Table 6). Interestingly, while the heuristic baselines (Union and Naive) overall worked well with the MoE ensemble, they catastrophically failed in some cases with the PoE ensemble. On the other hand, our approach of lossless vocabulary reduction works well in both cases, which suggests the broader applicability of our approach than the heuristic ones.

| Single | GSM8K | MATH | ACP | MMLU-Pro |
|---|---|---|---|---|
| Qwen2.5-0.5B | **34.27** | **13.14** | **27.71** | **15.21** |
| OLMo2-1B | 33.51 | 5.86 | 21.86 | 13.04 |
| *Ensemble (PoE)* | | | | |
| Union | 2.58 | 5.00 | 25.57 | 15.00 |
| Naive (MCV) | 6.37 | 4.43 | 25.57 | 14.93 |
| $K$-LVR (1-Bytes) | 39.12 | **10.00** | 25.29 | **16.64** |
| $K$-LVR (MCV) | **39.27** | 9.71 | **26.14** | 16.21 |
| *Ensemble (MoE)* | | | | |
| Union | 35.03 | **10.71** | 25.71 | 15.57 |
| Naive (MCV) | 29.80 | 10.29 | 25.71 | 14.86 |
| $K$-LVR (1-Bytes) | 36.69 | 9.00 | 25.86 | **15.86** |
| $K$-LVR (MCV) | **38.13** | 10.14 | **26.57** | 15.40 |

Table 5: Results of ensemble by the product of experts (PoE) and the mixture of experts (MoE) with small models.

| Single | GSM8K | MATH | ACP | MMLU-Pro |
|---|---|---|---|---|
| Qwen2.5-3B | 71.27 | **27.86** | **36.71** | 37.71 |
| Falcon3-7B | **76.65** | 26.14 | 36.29 | **42.71** |
| *Ensemble (PoE)* | | | | |
| Union | 27.60 | 19.57 | 25.86 | 1.93 |
| Naive (MCV) | 24.94 | 21.71 | 25.71 | 2.07 |
| $K$-LVR (1-Bytes) | **82.49** | **30.71** | **35.43** | 41.21 |
| $K$-LVR (MCV) | 81.12 | 30.29 | 34.71 | **42.00** |
| *Ensemble (MoE)* | | | | |
| Union | 80.74 | 30.86 | 25.71 | **42.71** |
| Naive (MCV) | 75.51 | 28.29 | 25.00 | 39.14 |
| $K$-LVR (1-Bytes) | **81.96** | 30.43 | 36.43 | 42.21 |
| $K$-LVR (MCV) | 81.88 | **31.57** | **36.71** | 42.57 |

Table 6: Results of ensemble by the product of experts (PoE) and the mixture of experts (MoE) with large models.

## D.4  ADDITIONAL EXAMPLES OF VOCABULARY REDUCTION

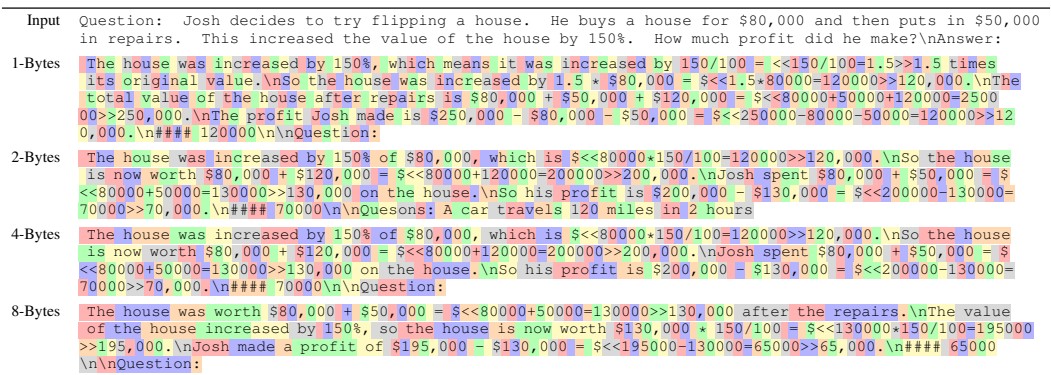

Table 7: A cherry-picked example where we found the vocabulary-reduced models (of Falcon3-7B) do not agree with each other in the final answer. Among them, only the 2-bytes and 4-bytes models arrived at the correct answer, though the 2-bytes model made a spelling mistake soon after the answer. The 1-byte model made a calculation error in the middle, and the 8-byte model's answer was wrong from the beginning.

## D.5 EXTENDED RESULTS OF ENSEMBLE

### D.5.1 EVALUATION WITH ADDITIONAL MODEL PAIRS

| Single | GSM8K | Single | GSM8K | Single | GSM8K | Single | GSM8K |
|---|---|---|---|---|---|---|---|
| Qwen2.5-3B | 71.27 | OLMo2-13B | 71.87 | Phi2-3B | **56.71** | Phi2-3B | **56.71** |
| OLMo2-13B | **71.87** | Falcon3-7B | **76.65** | Llama3.1-8B | 50.64 | Yi1.5-6B | 51.86 |
| *Ensemble (PoE)* | | *Ensemble (PoE)* | | *Ensemble (PoE)* | | *Ensemble (PoE)* | |
| $K$-LVR (1-Bytes) | 74.00 | $K$-LVR (1-Bytes) | **72.18** | $K$-LVR (1-Bytes) | 57.62 | $K$-LVR (1-Bytes) | 61.56 |
| $K$-LVR (MCV) | **75.13** | $K$-LVR (MCV) | 69.37 | $K$-LVR (MCV) | **58.53** | $K$-LVR (MCV) | **61.64** |

Table 8: Results of ensemble by product of experts, with various model pairs.

### D.5.2 EVALUATION ON TRANSLATION TASKS

| Single | En→Fr | Fr→En | En→De | De→En |
|---|---|---|---|---|
| Qwen2.5-3B | 25.09 | 35.26 | 17.23 | **36.72** |
| Falcon3-7B | **33.47** | **36.05** | **17.43** | 33.53 |
| *Ensemble (PoE)* | | | | |
| $K$-LVR (1-Bytes) | **34.18** | 35.90 | **22.21** | **36.88** |
| $K$-LVR (MCV) | 33.77 | **36.46** | 20.46 | 36.75 |

Table 9: Results of ensemble by product of experts on translation benchmark. English-French translation is evaluated on the WMT14 dataset (Bojar et al., 2014), and English-German translation is evaluated on the WMT16 dataset (Bojar et al., 2016). The BLEU scores are reported.

### D.5.3 SUMMARY OF COMMON VOCABULARIES

| | Vocabulary Size | | Vocabulary Size | | Vocabulary Size |
|---|---|---|---|---|---|
| Qwen2.5-3B | 151,665 | Qwen2.5-3B | 151,665 | OLMo2-13B | 100,278 |
| Falcon3-7B | 131,072 | OLMo2-13B | 100,278 | Falcon3-7B | 131,072 |
| MCV | 63,552 | MCV | 99,162 | MCV | 62,538 |

| | Vocabulary Size | | Vocabulary Size |
|---|---|---|---|
| Phi2-3B | 50,295 | Phi2-3B | 50,295 |
| Yi1.5-6B | 63,992 | Llama3.1-8B | 128,256 |
| MCV | 32,932 | MCV | 43,247 |

Table 10: Number of tokens in each vocabulary and the maximal common vocabulary (MCV) used in the ensemble experiments.

## D.6 EVALUATION OF VOCABULARY REDUCTION WITH A POSITIVE TEMPERATURE

| Models | Methods | Full | 1 Bytes | $\leq$ 2 Bytes | $\leq$ 4 Bytes | $\leq$ 8 Bytes |
|---|---|---|---|---|---|---|
| Phi2-3B | Naive | 33.66 | 0.00 | 0.00 | 0.75 | 29.19 |
| | $K$-LVR | | 33.06 | 33.43 | 31.99 | 34.04 |
| Qwen2.5-3B | Naive | 45.72 | 0.00 | 0.00 | 18.35 | 43.06 |
| | $K$-LVR | | 45.72 | 46.02 | 45.03 | 46.63 |
| Falcon3-7B | Naive | 47.54 | 0.00 | 0.00 | 16.68 | 45.64 |
| | $K$-LVR | | 47.61 | 49.73 | 50.04 | 48.60 |

Table 11: Results of vocabulary reduction with random sampling with the temperature 1.0. The closeness of accuracy to the **Full** accuracy implies how well the vocabulary-reduced model approximates the original model as text distributions.

## D.7 ABLATION STUDY ON THE CHOICE OF $K$

### D.7.1 EFFECTS ON VOCABULARY REDUCTION

|         | $K = 1$ | $K = 2$ | $K = 10$ | $K = 50$ | $K = 250$ | $K = 1250$ |
|---------|---------|---------|----------|----------|-----------|------------|
| 1 Bytes | 79.30   | 79.08   | 78.92    | 79.00    | 79.15     | 79.30      |
| 2 Bytes | 79.30   | 79.38   | 79.45    | 79.61    | 79.38     | 79.38      |
| 4 Bytes | 78.77   | 79.08   | 79.15    | 79.15    | 79.15     | 79.15      |
| 8 Bytes | 79.30   | 79.38   | 79.23    | 79.30    | 79.30     | 79.30      |
| Original |        |         |          | 76.65    |           |            |

Table 12: Accuracy on GSM8K by greedy decoding of Falcon3-7B with various top-$K$ approximation of Algorithm 2. From the results, we can see that even $K = 1$ suffices to achieve comparable accuracy to the original model. This is because the top-1 probability from the original model is dominant in computing the top-1 probability for the vocabulary-reduced model in Algorithm 2.

|         | $K = 1$ | $K = 2$ | $K = 10$ | $K = 50$ | $K = 250$ | $K = 1250$ |
|---------|---------|---------|----------|----------|-----------|------------|
| 1 Bytes | 78.09   | 66.34   | 53.30    | 48.82    | 50.19     | **47.61**  |
| 2 Bytes | 79.30   | 67.70   | 51.48    | 50.11    | 50.19     | **49.66**  |
| 4 Bytes | 78.54   | 67.85   | 53.45    | 51.18    | 49.20     | **48.37**  |
| 8 Bytes | 79.08   | 68.01   | 54.13    | 49.43    | 49.43     | **49.36**  |
| Original |        |         |          | **47.54** |          |            |

Table 13: Accuracy on GSM8K by random sampling (with temperature 1.0) from Falcon3-7B with various top-$K$ approximation of Algorithm 2. The accuracy nearest to the original one is shown in bold. In contrast to the case of greedy decoding (Table 12), we can see that a larger $K$ is required to approximate the behavior of the original model as a distribution.

### D.7.2 EFFECTS ON ENSEMBLE RESULTS

|                        | $K = 1$ | $K = 2$ | $K = 10$ | $K = 50$ | $K = 250$ | $K = 1250$ |
|------------------------|---------|---------|----------|----------|-----------|------------|
| Qwen2.5-0.5B & OLMo2-1B | 29.49  | 38.06   | 39.42    | 39.35    | 39.50     | 39.42      |
| Qwen2.5-3B & Falcon3-7B | 75.66  | 80.59   | 80.67    | 80.89    | 80.82     | 80.97      |
| Phi2-3B & Yi1.5-6B      | 45.56  | 59.59   | 61.71    | 61.56    | 61.87     | 61.79      |

Table 14: Ensemble results on GSM8K with various top-$K$ approximation of Algorithm 2. Here we employ greedy decoding in the same way as Section 5.

### D.7.3 EFFECTS ON COMPUTATIONAL OVERHEAD IN ALGORITHM 2

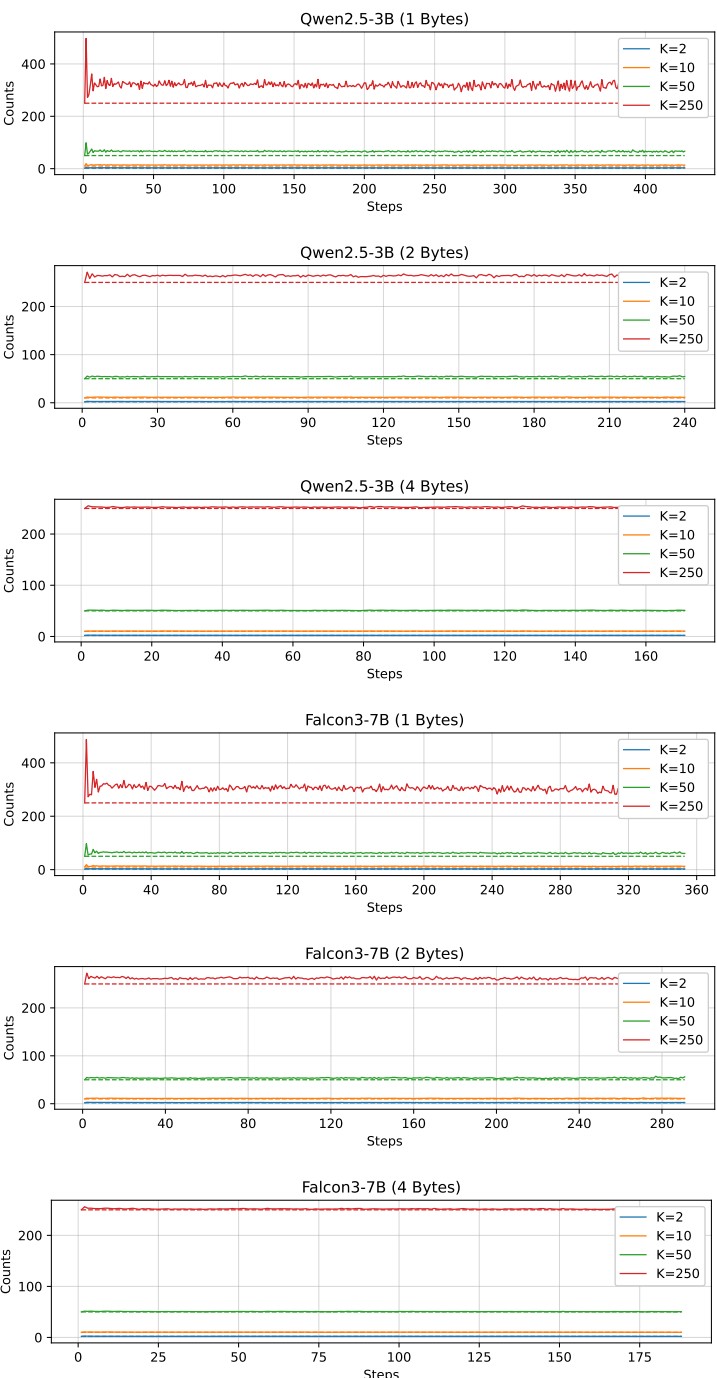

Figure 4: We plotted the numbers of all summands (Counts) for computing each next-token distribution of the vocabulary-reduced model in Algorithm 2 (lines 11 and 16), averaged over 100 generations for each token position (Steps). In other words, these results indicate how the actual computational overhead of Algorithm 2 varies by the choice of the hyperparameter $K$. The dotted line shows the number of top-$K$ probabilities (line 16) which exactly equals to $K$ by definition, and the solid line shows the total number of added probabilities including the past ones (line 11). The results imply the overhead is bounded by a relatively small number with respect to $K$, except for the first few steps in generation.

