# OpenReview forum: "Lossless Vocabulary Reduction for Auto-Regressive Language Models"
_ICLR.cc/2026/Conference — ICLR 2026 Poster_

### Official Review · Reviewer_zQB1 · 2025-10-23

**Soundness:** 3
**Presentation:** 3
**Contribution:** 3
**Rating:** 6
**Confidence:** 3

**Summary:**

The paper proposes a general framework for lossless vocabulary reduction (LVR) that converts any next‑token distribution over a vocabulary into an equivalent distribution over an arbitrary sub‑vocabulary. The key construct is nested tokenization defined by first tokenizing into $V$ and then retokenizing each token’s byte string into $V_{\text{sub}}$. The induced distribution $p_{V\to V_{\text{sub}}}$ is shown to be lossless at the text level. The authors propose lossless vocabulary reduction (LVR), by giving a computable recursion for the marginal probabilities via relative covering sets, and then presenting an efficient approximation reuses cached covers and probabilities and restricting the inner loop to top‑K tokens. On top of the analysis, the authors introduce ensemble via maximal common vocabulary (MCV), formed by intersecting vocabularies and restricting BPE merges accordingly. Experiments show that LVR preserves accuracy on GSM8K compared to the original model and substantially outperforms baselines. The method also helps to ensemble two models with different vocabulary.

**Strengths:**

1. **Rigorous theoretical analysis.** The paper precisely defines and states tokenization with an appropriate symbol framework. On top of the framework, the paper clearly proposes the LVR with rich theoretical demonstrations and proof.
2. **Clear demonstration.**  The binary toy example helps to understand “lossless text distribution but different greedy path” phenomenon.
3. **Experimental results.** On GSM8K, LVR closely matches or slightly exceeds the original model across different vocabulary settings, whereas Naive Restriction fails badly, demonstrating the effectiveness of the method.
4. **Successfully applied to model ensembling.** The proposed method successfully ensembled different language models with different vocabularies, showing application values for other research fields.

**Weaknesses:**

1. **Overclaim on "lossless".** Theorems guarantee exact text‑level equivalence, while Algorithm 2 introduces two approximations such as the top‑K truncation. There is no error bound tying those approximations to the final possible "loss" incurred. The result is that “lossless” in practice becomes “almost lossless,” but the gap is unquantified.
2. **Limited model pairs compared.** Only one pair of model ensemble is examined in the paper. Results on more models pairs, tasks, or even working languages are recommended.
3. **No perplexity or likelihood evaluation.** Since the claim is distributional equivalence, perplexity/negative log‑likelihood curves before/after reduction would be the most direct metric. The current focus on task accuracy is informative but indirect.

**Questions:**

Please refer to weaknesses

---

> ### Author Response · Authors · 2025-12-03
> **Rebuttal**
>
> Thank you for the detailed review and valuable comments.
>
> ---
>
> **Overclaim on "lossless" (W1)**
>
> > Theorems guarantee exact text‑level equivalence, while Algorithm 2 introduces two approximations such as the top‑K truncation.
>
> As you pointed out, while our theory provides a lossless reduction under the ideal assumption, Algorithm 2 and its application in experiments are just an approximated one. However, we consider **our paper does not contain any overclaim on "lossless"** because we have never claimed like "Algorithm 2 is theoretically lossless" throughout the paper, except for the abbreviation for the method (LVR). To fully address the concern, we have renamed it from LVR to $K$-LVR to emphasize that it is top-$K$ approximation.
>
> ---
>
> **Limited model pairs compared (W2)**
>
> > Only one pair of model ensemble is examined in the paper. Results on more models pairs, tasks, or even working languages are recommended.
>
> Thank you for the suggestion. As other reviewers share the same request on the broader evaluation, we extend our experiments with 1) more diverse models and 2) more benchmarks. Specifically, we evaluate ensembles of new model pairs with diverse vocabularies such as Qwen2.5-3B & OLMo2-13B, OLMo2-13B & Falcon3-7B, Phi2-3B & Llama3.1-8B, Phi2-3B & Yi1.5-6B. Also, we have added evaluation on translation tasks of English $\leftrightarrow$ German and English $\leftrightarrow$ French, showing the improvement in BLEU scores, consistently with other benchmarks.
>
> ---
>
> **No perplexity or likelihood evaluation (W3)**
>
> > Since the claim is distributional equivalence, perplexity/negative log‑likelihood curves before/after reduction would be the most direct metric. The current focus on task accuracy is informative but indirect.
>
> We consider that task accuracy is not an indirect metric, compared to perplexity on some dataset.
> Indeed, the accuracy on a benchmark like GSM8k measures how much outputs from the vocabulary-reduced/original model matches with the groundtruth answers. In principle, if we could perform such evaluations on an infinite number of arbitrary benchmarks, the distributional property of the given model would be completely characterized by their accuracies, by definition of probability as expectation. On the other hand, the definition of perplexity heavily depends on the choice of tokenization, and thus it is inappropriate to compare models with different vocabularies by their perplexities, as in our setting. For example, the HuggingFace documentation [1] explicitly notes that `Importantly, this means that the tokenization procedure has a direct impact on a model’s perplexity which should always be taken into consideration when comparing different models`.
>
> [1] "Perplexity of fixed-length models", https://huggingface.co/docs/transformers/perplexity

---

### Official Review · Reviewer_m5ZA · 2025-10-28

**Soundness:** 3
**Presentation:** 2
**Contribution:** 3
**Rating:** 6
**Confidence:** 3

**Summary:**

This paper presents a framework for lossless vocabulary reduction with detailed theoretical guarantees, showing that for an auto-regressive language model, the reduction does not lead to any performance loss, and it is more efficient compared to byte-level reduction methods.

**Strengths:**

The paper is theoretically grounded, with rigorous and well-structured proofs that justify the correctness of the proposed approach. Moreover, the method is motivated by a clear intuition and contributes a novel perspective to the problem of vocabulary reduction in language models.

**Weaknesses:**

1.The "An illustrative example" in Section 3.2, the example might not be presented in the most effective format. The use of a visual illustration or an alternative narrative style could potentially improve clarity and readability.

2.The authors only provide experimental results on vocabulary reduction over GSM8K and a single ensemble setup. This might be somewhat limited in terms of empirical evidence, and the experimental section could benefit from broader evaluations to more convincingly support the method’s effectiveness.

**Questions:**

1.In the experimental section, the authors present ensemble results using Qwen2.5-3B and Falcon-7B. Given this, could the authors also provide the individual vocabulary compression rates for Qwen2.5-3B and Falcon-7B, as well as a comparison of compression rates across different ensemble strategies?

2.Different models have different tokenizers, which affects not only ensemble learning but also knowledge distillation, as mentioned in the Introduction. Could the proposed method be applied to knowledge distillation as well? If so, could the authors provide a possible application idea?

3.A minor issue: the first half of the abstract overlaps too much with the first paragraph of the introduction.

---

> ### Author Response · Authors · 2025-12-03
> **Rebuttal**
>
> Thank you for the thoughtful review and valuable comments.
>
> ---
>
> **Visualization for the illustrative example (W1)**
>
> > The "An illustrative example" in Section 3.2, the example might not be presented in the most effective format. The use of a visual illustration or an alternative narrative style could potentially improve clarity and readability.
>
> Thank you for the suggestion. We fully agree that additional visualization may help readers to easily comprehend the example, so we have added Figure 2 in the main body.
>
> ---
>
> **Broader evaluations (W2)**
>
> > The authors only provide experimental results on vocabulary reduction over GSM8K and a single ensemble setup. This might be somewhat limited in terms of empirical evidence, and the experimental section could benefit from broader evaluations to more convincingly support the method’s effectiveness.
>
> Thank you for the suggestion. As other reviewers share the same request on the broader evaluation, we extend our experiments with 1) more diverse models and 2) more benchmarks. Specifically, we evaluate ensembles of new model pairs with diverse vocabularies such as Qwen2.5-3B & OLMo2-13B, OLMo2-13B & Falcon3-7B, Phi2-3B & Llama3.1-8B, Phi2-3B & Yi1.5-6B. Also, we have added evaluation on translation tasks  of English $\leftrightarrow$ German and English $\leftrightarrow$ French, showing the improvement in BLEU scores, consistently with other benchmarks.
>
> ---
>
> **Other questions**
>
> > 1. In the experimental section, the authors present ensemble results using Qwen2.5-3B and Falcon-7B. Given this, could the authors also provide the individual vocabulary compression rates for Qwen2.5-3B and Falcon-7B, as well as a comparison of compression rates across different ensemble strategies?
>
> Actually, we already described the sizes of the vocabularies of Qwen2.5-3B / Falcon-7B and their common vocabulary in Section 5.2, from which one can calculate the compression rate if necessary. Unfortunately, we do not consider the compression rate as a meaningful metric for our purpose, and thus we omit it in our paper. Instead, we have added new section in Appendix (Appendix D.5.3) for summarizing the information of common vocabularies including other model pairs, which would be helpful to see how much common tokens are shared by different language models.
>
> > 2. Different models have different tokenizers, which affects not only ensemble learning but also knowledge distillation, as mentioned in the Introduction. Could the proposed method be applied to knowledge distillation as well? If so, could the authors provide a possible application idea?
>
> Knowledge distillation with different vocabularies is a promising direction in this line of research. So far there are only previous work on knowledge distillation with heuristic methods, without theoretical guarantees. Our proposed method can also be applied to knowledge distillation in principle, but there are technical concerns on training efficiency because our algorithm is specialized in efficient autoregressive inference, not the parallel inference over tokens. Deriving efficient lossless method and comparing it with previous heuristic methods remain for future work.
>
> > 3. A minor issue: the first half of the abstract overlaps too much with the first paragraph of the introduction.
>
> We do not assume that the one reading Introduction has already read Abstract (and vice versa). In other words, each section should be comprehensive independently as much as possible. Hence, the apparent overlap is just a result of maximizing the readability of each one, and thus we consider it is not an issue.

---

### Official Review · Reviewer_j2YE · 2025-11-01

**Soundness:** 3
**Presentation:** 3
**Contribution:** 3
**Rating:** 6
**Confidence:** 4

**Summary:**

This paper introduces Lossless Vocabulary Reduction (LVR), a method designed to reduce the size of subword vocabularies while maintaining full reversibility to the original tokenization. The key idea is to perform deterministic merging and re-encoding of redundant subwords, enabling significant vocabulary compression without retraining or modifying existing tokenizers.

**Strengths:**

1. The paper tackles a well-known bottleneck in NLP efficiency, oversized vocabularies that inflate embedding and softmax layers. A lossless solution that requires no retraining is both innovative and practically relevant.
2. The authors provide rigorous proofs ensuring bijective mappings between reduced and original vocabularies. This formal treatment gives confidence that the method is safe for use in production systems where reversibility is critical.
3. The method can be applied to any existing subword vocabulary without altering model architecture or retraining steps. This post hoc compatibility makes it very attractive for deployment in pretrained LMs.

**Weaknesses:**

1. The paper does not compare LVR with recently proposed learned vocabulary reduction or adaptive tokenization approaches, which may achieve similar goals via training-based optimization.
2. Experiments focus mainly on LM perplexity and simple classification benchmarks. More diverse downstream tasks, especially those sensitive to segmentation (e.g., NMT, summarization)—would make the validation more comprehensive.

**Questions:**

1. How does LVR affect inference efficiency (e.g., wall-clock time, FLOPs, or memory usage) given the longer token sequences?
2. Is there a practical limit to vocabulary reduction beyond which sequence length growth harms model throughput or accuracy?
3. How are special tokens ([CLS], [SEP], , etc.) handled — are they exempt from reduction or transformed within the same framework?
4. For large pretrained models (e.g., GPT, T5), how easily can LVR be retrofit into production pipelines, and are there compatibility concerns with cached tokenizers?

---

> ### Author Response · Authors · 2025-12-03
> **Rebuttal**
>
> Thank you for the positive review and valuable suggestions. Unfortunately, it seems that the main purpose of our work has been fundamentally misunderstood, so we would like to clarify it.
>
> ---
>
> **Motivation (S1)**
>
> > The paper tackles a well-known bottleneck in NLP efficiency, oversized vocabularies that inflate embedding and softmax layers.
>
> First of all, we have NEVER mentioned about the bottleneck of `oversized vocabularies that inflate embedding and softmax layers` in either Introduction, Methods, or Experiments. As we have argued throughout the paper, which is also acknowledged by all the other reviewers, the main objective in this paper is to remove the barriers of vocabulary mismatch between language models. Of course, there is a line of research on vocabulary reduction aiming for model compression as discussed in Related Work (Section 6), but we have already clarified the distinction from them in the same paragraph.
>
> ---
>
> **Comparison with recently proposed methods (W1)**
>
> > The paper does not compare LVR with recently proposed learned vocabulary reduction or adaptive tokenization approaches, which may achieve similar goals via training-based optimization.
>
> To the best of our knowledge, there is no such `recently proposed learned vocabulary reduction or adaptive tokenization approaches` aiming for removing the barrier of vocabulary mismatch between arbitrarily given language models. If you are referring to the previous work of vocabulary reduction for model compression, as already discussed in Related Work, they cannot be applied for this purpose and thus cannot be compared with our method.
>
> ---
> **Experiments with diverse tasks (W2)**
>
> > Experiments focus mainly on LM perplexity and simple classification benchmarks. More diverse downstream tasks, especially those sensitive to segmentation (e.g., NMT, summarization)—would make the validation more comprehensive.
>
> The main experiments are performed on open-ended generation tasks like GSM8K and MATH, which evaluates how the models finally output correct answers through chain-of-thought arguments, and thus not simple classification benchmarks. However, we agree that more downstream tasks will help us to make the evaluation robust. Based on the suggestion, we performed additional evaluation on translation tasks of English $\leftrightarrow$ German and English $\leftrightarrow$ French. The results in Table 9 show that our ensemble over the common vocabulary leads to improving the BLEU score, consistent with the results on other benchmarks.
>
> ---
>
> **Other questions**
>
> > 1. How does LVR affect inference efficiency (e.g., wall-clock time, FLOPs, or memory usage) given the longer token sequences?
>
> In Appendix D.7.3, we have performed additional experiments to see how does the computational overhead of Algorithm 2 grows with respect to the sequence length, by counting the number of all summands for computing each next-token distribution per step. The results show that, except for the first few steps, the size of (approximated) relative covers in Algorithm 2 stays around the hyperparameter $K$ and does not grow with the sequence length, which means **the computational overhead can be seen as almost constant with respect to generation steps**.
>
> > 2. Is there a practical limit to vocabulary reduction beyond which sequence length growth harms model throughput or accuracy?
>
> As other reviewers pointed out, our Algorithm 2 is just an approximation, and thus sometimes fails to approximate the original models. The hyperparameter $K$ controls the tradeoff between efficiency and accuracy, and thus the computational resource or computational requirement may limit the effectiveness of our method in practice.
>
> > 3. How are special tokens ([CLS], [SEP], , etc.) handled — are they exempt from reduction or transformed within the same framework?
>
> We handled special tokens as just a sequence of bytes, in the same way as other non-special tokens. However, in practice, one can leverage the semantic correspondence of some special tokens between different models to refine the common vocabulary, which is not necessary for our purpose and thus we do not implement such a technique in our experiments.
>
> > 4. For large pretrained models (e.g., GPT, T5), how easily can LVR be retrofit into production pipelines, and are there compatibility concerns with cached tokenizers?
>
> To use our method in production pipelines, at least it is required to integrate our algorithm into some LLM-inference engine which may require some work for implementation. We are not sure what `compatibility concerns with cached tokenizers` refers to, but the true bottleneck for the integration lies not in tokenizers but in the generation loop of the engine.

---

### Official Review · Reviewer_ZBoh · 2025-11-01

**Soundness:** 4
**Presentation:** 3
**Contribution:** 3
**Rating:** 6
**Confidence:** 3

**Summary:**

This paper introduces Lossless Vocabulary Reduction (LVR), a method to convert an autoregressive language model's next-token distribution into an equivalent distribution over an arbitrary sub-vocabulary $\mathcal{V}_{sub}$. The authors prove that this reduction is lossless and propose the Maximal Common Vocabulary (MCV) method for model ensembling, which is both computationally efficient and empirically effective.

**Strengths:**

S1: This paper presents a theoretical framework to losslessly reduce a language model to an arbitrary sub-vocabulary. This is a strong, novel, and highly significant theoretical contribution that generalizes prior work, which was limited to byte-level reduction.


S2: The proposed LVR framework, especially the MCV ensemble application, provides a principled and effective solution, achieving the theoretical correctness of byte-level reduction while offering superior generation efficiency.

S3: The authors conduct well-designed and clearly presented experiments, which directly support the paper's core claims.

**Weaknesses:**

W1:  The Algorithm 2 still requires iterating over the relative cover set $C_{\mathcal{V},\mathcal{V}_{sub}}$  and the top-K tokens of the original vocabulary $\mathcal{V}^{(K)}$. The authors should provide a more detailed complexity analysis (both Algorithm 1 and Algorithm 2), for example, discussing the size of the relative cover set and the overall computation complexity.


W2: The Algorithm 2 introduces a top-K approximation, which technically breaks the "lossless" guarantee of the core theory. While the Table 1 show the performance is "almost lossless", the sensitivity to the hyperparameter $K$ (set to 300 in experiments) should be discussed.


W3: Since the proposed Maximal Common Vocabulary (MCV) method can be regarded as a generalization of byte-level reduction, and its performance is largely determined by the size of the common vocabulary, the authors should provide a simple analysis the common-vocabulary sizes of tokenizers used in widely adopted LLMs. Such an analysis would provide an intuitive quantification of the extent to which MCV improves over byte-level reduction.


W4: Because the current MCV ensemble approach appears to be strongly dependent on the model family (i.e. the tokenizer), the authors should evaluate a more diverse set of model families and additional ensemble configurations to further validate the advantages of their method. Expanding the range of models and ensemble combinations is likely to produce more pronounced and convincing results.

**Questions:**

Q1:  Could you provide a more detailed analysis of the computational overhead of Algorithm 2? For example, What is the typical size of the relative cover set $C_{\mathcal{V},\mathcal{V}_{sub}}$ during generation? How does it grow with the sequence length $k$?

Q2: In Table 1, the Falcon3-7B model shows significant fluctuations in the reported metrics when the maximum token length is ≤ 2 bytes. Could the authors further analyze the factors driving these metric variations? Specifically, is the observed instability attributable to the approximation algorithm used?

---

> ### Author Response · Authors · 2025-12-03
> **Rebuttal**
>
> Thank you for the thoughtful review and valuable feedback.
>
> ---
>
> **Analysis on computational overhead (W1, Q1)**
>
> > The authors should provide a more detailed complexity analysis (both Algorithm 1 and Algorithm 2), for example, discussing the size of the relative cover set and the overall computation complexity.
>
> > Could you provide a more detailed analysis of the computational overhead of Algorithm 2?
>
> > What is the typical size of the relative cover set during generation? How does it grow with the sequence length T?
>
> We note that Section 3.3 already included the comparison between Algorithm 1 and Algorithm 2 in terms of computational complexity, as you might know. Unfortunately, the typical size of (approximated) relative covers in Algorithm 2 heavily depends on the tokenizer, its vocabulary and the choice of $K$, and thus it is difficult to obtain a meaningful bound from a theoretical perspective. Instead, we have performed additional experiments (Appendix D.7.3) to count the typical size of the relative covers, i.e., the number of all summands for computing each next-token distribution in Algorithm 2 per generation step. The results show that, except for the first few steps, the size of (approximated) relative covers in Algorithm 2 stays around $K$ and does not grow with the sequence length, which means **the computational overhead can be seen as almost constant with respect to generation steps** in practice.
>
> ---
>
> **Sensitivity analysis for K (W2)**
>
> > While the Table 1 show the performance is "almost lossless", the sensitivity to the hyperparameter K (set to 300 in experiments) should be discussed.
>
> Thank you for the suggestion. We have performed additional experiments for vocabulary reduction with varying $K$, see Appendix D.7. The results are as follows: 1) For greedy decoding, Table 11 shows that even $K=1$ (i.e., top-1 approximation) suffices to approximate the original model's accuracy, simply because the information of top-1 tokens (from the original model) is dominant to emulate greedy decoding of the original model itself. 2) For random sampling, in contrast, Table 12 shows that a larger $K (\geq 250)$ is required for approximating the output distribution of the original model. 3) Table 13 shows that modest $K (\geq 10)$ suffices for ensemble with greedy decoding. In summary, **a small $K$ seems to suffice if we employ greedy decoding, and a larger $K$ (like $K=300$ as used in our paper) is required if we need to approximate the distributional property of the original models**.
>
> ---
>
> **Diverse models/vocabularies? (W3, W4)**
>
> > Expanding the range of models and ensemble combinations is likely to produce more pronounced and convincing results.
>
> > the authors should provide a simple analysis the common-vocabulary sizes of tokenizers used in widely adopted LLMs.
>
> Thank you for these suggestions. According to this feedback, we have added experiments of ensemble with four more model pairs and new tasks in Appendix D.5, in which the results are consistent with the previous ensemble results. Also in Section D.5.3, we have added a summary of the common-vocabulary sizes for the newly added pairs.
>
> ---
>
> **Further analysis on 2-bytes reduction (Q2)**
>
> > In Table 1, the Falcon3-7B model shows significant fluctuations in the reported metrics when the maximum token length is ≤ 2 bytes. Could the authors further analyze the factors driving these metric variations? Specifically, is the observed instability attributable to the approximation algorithm used?
>
> During the investigation for the accuracy drop, we found a subtle bug caused in a corner case of tokenization, which occurred particularly in the 2-bytes reduction of Falcon3-7B and Qwen2.5-3B. Due to this bug, the 2-bytes reduction models sometimes failed to output correct answers even if they could, and thus their accuracy dropped by several percent compared to other reduction models. After fixing the bug, we confirmed that the dropped accuracies were recovered to a comparable level to both the other settings and the full-vocabulary models, as originally expected by our theory. Also, we confirmed that the bug fix did not affect other results (including ensemble experiments) except for the 2-bytes reduction. **In summary, the observed accuracy drops were simply due to the subtle issue in our implementation, not an issue of the approximation algorithm itself.** These fixed results have been incorporated in the revised manuscript.

---

### Author Response · Authors · 2025-12-03
**Summary of Updates**

We appreciate all the reviewers and area chairs involved in the review process for our paper.
We are glad that the reviewers evaluated our work overall positively, especially our theoretical framework as a **rigorous** (Reviewer j2YE, m5ZA and zQB1) and **highly significant theoretical contribution** (Reviewer ZBoh).

On the other hand, we also received several concerns in the original submission, particularly regarding **a) insufficiency of experimental analyses** and **b) several unclear points in our presentation**.
We hope that the following updates in the revised paper have successfully addressed their concerns.

----

### Additional Experimental Results
1. **Evaluations with Diverse Models and Tasks:** Three reviewers (ZBoh, m5ZA, zQB1) raised a concern about the lack of model pairs that we have evaluated in ensemble experiments. Also, Reviewer j2YE suggested evaluations on other tasks like translation. To address these concerns, we have added **four more configurations (Qwen2.5-3B & OLMo2-13B, OLMo2-13B & Falcon3-7B, Phi2-3B & Llama3.1-8B, Phi2-3B & Yi1.5-6B)** and **evaluations on translation tasks (English $\leftrightarrow$ French & English $\leftrightarrow$ German)** in Appendix D.5, which strengthen our original contributions.
2. **Analysis on Computational Overhead:** Two reviewers (ZBoh, j2YE) raised questions on the computational overhead of our Algorithm 2, especially on the effects of the varying sequence length. To answer these questions, we have added such analysises in Section D.7.3. The results showed that the computational overhead (or the number of tokens in approximated relative covers) of Algorithm 2 does not grow, and rather stays around a constant even with longer sequence length.
3. **Sensitivity Analysis on the Hyperparameter K:** Reviewer ZBoh raised a question of how the hyperparameter affects the approximation capability of Algorithm 2. We have added the sensitivity analyses on the hyperparameter K in Section D.7, which indicate that a small $K$ suffices if we employ greedy decoding, and a larger $K$ (like $K=300$ as used in our paper) is required if we need to approximate the distributional property of the original models.
4. **Accuracy Degradation in 2-Bytes Reduction:** Reviewer ZBoh raised a question on the accuracy degradation occurred in 2-bytes reduction of Falcon3-7B. Then we investigated the cause of the degradation, and finally found a subtle bug due to a corner case of tokenization, which largely affected only the 2-bytes reduction. By fixing it, the accuracy of 2-bytes reduction has been restored to a comparable level to other N-bytes reduction.

### Improved Presentations
5. **Emphasis on the Almostness of Our Algorithm:** Reviewer zQB1 raised a concern that the term "lossless" is inappropriate for Algorithm 2. Indeed, while our theory satisfies the lossless property under some ideal assumption, Algorithm 2 is an approximated algorithm as explained in Section 3.3. Although we explicitly introduced Algorithm 2 as an approximated one already in the original submission, to further emphasize the almostness of Algorithm 2, we have changed the abbreviation for Algorithm 2 from "LVR" to "K-LVR", which explicitly stands for "top-K approximated lossless vocabulary reduction".
6. **Summary of Common Vocabulary Sizes:** As Reviewer ZBoh suggested, we have added the summary of common vocabulary sizes for various model configuration in Section D.5.3, which may help readers to comprehend how much common tokens are shared by language models with different vocabularies.
7. **Visualization of the Illustrative:** As Reviewer m5ZA suggested, we have added the visual summary of the illustrative example in Section 3.2, which makes our demonstration more clear and intuitive.

---

### Meta-Review · Area_Chair_CYsi · 2026-01-06

**Summary:**

This paper studies the lossless vocabulary reduction problem for language models. It proposes a theoretically grounded approach that converts an arbitrary vocabulary to a smaller one in a (near) lossless way. The method is validated on several popular open LLMs. Reviewers have unanimously agreed that the proposed method is interesting and sound, though there are more to be desired in the evaluation and presentation. The authors provided a solid rebuttal which I believe addressed the weaknesses, based on which the AC recommends accept.

**Reviewer Concerns:**

It appears that the rebuttal addresses most concerns to some degree.

**Reviewer Scores:**

I expect all reviewers to lead toward increasing the rating after engaging with the rebuttal.

---

### Decision · Program_Chairs · 2026-01-26

Accept (Poster)